# Local retinoic acid signaling directs emergence of the extraocular muscle functional unit

Glenda Evangelina Comai[1,2]*, Markéta Tesařová[3], Valérie Dupé[4], Muriel Rhinn[5], Pedro Vallecillo-García[6], Fabio da Silva[7,8], Betty Feret[5], Katherine Exelby[9], Pascal Dollé[5], Leif Carlsson[10], Brian Pryce[11], François Spitz[12], Sigmar Stricker[6], Tomáš Zikmund[3], Jozef Kaiser[3], James Briscoe[9], Andreas Schedl[7], Norbert B. Ghyselinck[5], Ronen Schweitzer[11], Shahragim Tajbakhsh[1,2]*

1 Stem Cells & Development Unit, Institut Pasteur, Paris, France, 2 CNRS UMR 3738, Institut Pasteur, Paris, France, 3 Central European Institute of Technology, Brno University of Technology, Brno, Czech Republic, 4 Université de Rennes, CNRS, IGDR, Rennes, France, 5 IGBMC-Institut de Génétique et de Biologie Moleculaire et Cellulaire, Illkirch, France, 6 Institute for Chemistry and Biochemistry, Freie Universität Berlin, Berlin, Germany, 7 Université Côte d'Azur, INSERM, CNRS, iBV, Nice, France, 8 Division of Molecular Embryology, German Cancer Research Center (DKFZ), Heidelberg, Germany, 9 The Francis Crick Institute, London, United Kingdom, 10 Umeå Center for Molecular Medicine, Umeå University, Umeå, Sweden, 11 Research Division, Shriners Hospital for Children, Portland, United States of America, 12 Genomics of Animal Development Unit, Institut Pasteur, Paris, France

* comai@pasteur.fr (GEC); shaht@pasteur.fr (ST)

**Data Availability Statement:** All relevant data are within the paper and its Supporting Information files.

## Abstract

Coordinated development of muscles, tendons, and their attachment sites ensures emergence of functional musculoskeletal units that are adapted to diverse anatomical demands among different species. How these different tissues are patterned and functionally assembled during embryogenesis is poorly understood. Here, we investigated the morphogenesis of extraocular muscles (EOMs), an evolutionary conserved cranial muscle group that is crucial for the coordinated movement of the eyeballs and for visual acuity. By means of lineage analysis, we redefined the cellular origins of periocular connective tissues interacting with the EOMs, which do not arise exclusively from neural crest mesenchyme as previously thought. Using 3D imaging approaches, we established an integrative blueprint for the EOM functional unit. By doing so, we identified a developmental time window in which individual EOMs emerge from a unique muscle anlage and establish insertions in the sclera, which sets these muscles apart from classical muscle-to-bone type of insertions. Further, we demonstrate that the eyeballs are a source of diffusible all-trans retinoic acid (ATRA) that allow their targeting by the EOMs in a temporal and dose-dependent manner. Using genetically modified mice and inhibitor treatments, we find that endogenous local variations in the concentration of retinoids contribute to the establishment of tendon condensations and attachment sites that precede the initiation of muscle patterning. Collectively, our results highlight how global and site-specific programs are deployed for the assembly of muscle functional units with precise definition of muscle shapes and topographical wiring of their tendon attachments.

**Funding:** We acknowledge funding support from the Institut Pasteur, Association Française contre le Myopathies, Agence Nationale de la Recherche (Laboratoire d'Excellence Revive, Investissement d'Avenir; ANR-10-LABX-73) and the Centre National de la Recherche Scientifique. We gratefully acknowledge the UtechS Photonic BioImaging (Imagopole), C2RT, Institut Pasteur, supported by the French National Research Agency (France BioImaging; ANR–10–INSB–04; Investments for the Future). MT, TZ and JK acknowledge the project CEITEC 2020 (LQ1601) with financial support from the Ministry of Education, Youth and Sports of the Czech Republic under the National Sustainability Programme II and Ceitec Nano+ project CZ.02.01/0.0./.0.0./16_013/0001728 under the program OP RDE. MT was financially supported by by the Brno City Municipality as a Brno Ph.D. Talent Scholarship Holder. The funders had no role in study design, data collection and analysis, decision to publish, or preparation of the manuscript.

**Competing interests:** The authors have declared that no competing interests exist.

**Abbreviations:** ADH, alcohol dehydrogenase; ATRA, all-trans retinoic acid; ALDH1A1, Aldehyde Dehydrogenase 1 Family Member A1; ALDH1A3, Aldehyde Dehydrogenase 1 Family Member A3; E, embryonic day; EOM, extraocular muscle; GFP, green fluorescent protein; KO, knock-out; LHX2, LIM Homeobox 2; Mesp1, mesoderm posterior BHLH transcription factor 1; mGFP, membrane tagged GFP; micro-CT, micro-computed tomography; MYF5, myogenic factor 5; MYOD, myogenic differentiation 1; MYOG, Myogenin; MyHC, myosin heavy chain; NCC, neural crest cell; PAX6, Paired Box 6; PAX7, Paired Box 7; PITX2, Paired-like homeodomain transcription factor 2; POM, periocular mesenchyme; RAR, retinoic acid receptor; RARE, retinoic acid response element, RDH, retinol dehydrogenase; RPE, retinal pigmented epithelium; RXR, retinoid X receptor; Scx, scleraxis; Shh, Sonic Hedgehog; SMA, alpha smooth muscle actin; SOX9, SRY-box containing gene 9; TCF4, Transcription factor 4; Tom, tdTomato; WMIF, whole-mount immunofluorescence; Wnt1, Wnt Family Member 1.

## Introduction

Acquisition of shape and pattern during development depends on the orchestrated crosstalk between a variety of tissues and cell types. Although significant knowledge on the mechanisms of differentiation and patterning within individual tissues has been attained, much less is known on how patterning of different adjacent tissues is integrated. The vertebrate musculoskeletal system serves as an ideal model to study these processes as different tissues including muscle, tendon and their attachments need to be articulated in 3D for proper function [1,2].

Among the craniofacial muscles, the morphological configuration of the extraocular muscles (EOMs) has been a longstanding challenge in comparative anatomy and evolutionary biology. Besides specialized adaptations, the basic EOM pattern is shared among all vertebrate classes [3–5] and includes 4 recti muscles (the superior rectus, the medial rectus, the inferior rectus, and the lateral rectus) and 2 oblique muscles (superior oblique and inferior oblique) for movement of the eyeball. Most vertebrates have also accessory ocular muscles that serve to retract the eye (the retractor bulbi) or control eyelid elevation (the levator palpabrae superioris) [3,6]. As such, the EOMs constitute an archetypal and autonomous functional unit for the study of how muscles, tendons, and tendon attachments are integrated with the development of the eyeball, their target organ.

Craniofacial muscles are derived from cranial paraxial and prechordal head mesoderm [3,7]. The corresponding connective tissues, i.e., tendons, bones, cartilages, and muscle connective tissue, were reported to be derived from cranial neural crest cells (NCCs) [7,8]. Although early myogenesis is NCC independent, NCCs later regulate the differentiation and segregation of muscle precursors, dictate the pattern of muscle fiber alignment, and that of associated skeletal and tendon structures [9–14]. Moreover, deletion of several genes in NCCs demonstrated their non-cell-autonomous roles in muscle morphogenesis at the level of the jaw [12,15,16], extraocular [17,18], and somitic-derived tongue muscles [19]. However, the full series of events driving morphogenesis of craniofacial musculoskeletal functional units is unexplored to date, in part because of the anatomical complexity of their configuration in the head. Moreover, understanding the developmental mechanisms that allow musculoskeletal connectivity is essential to understand the anatomical diversification that took place during the evolution of the vertebrate head. Yet, proximate factors that allow cross-tissue communication for coordinated emergence of the individual muscle masses with that of their tendons and attachment sites are poorly defined.

Much of our understanding of musculoskeletal development and integration into functional units comes from studies in the limb. Lateral plate mesoderm-derived muscle connective tissue cells and tendon primordia establish a pre-pattern that determines the sites of myogenic differentiation and participate in splitting of the muscle masses in the limb [20–22]. Tendons connect muscles to the skeleton and are formed by scleraxis (Scx)-expressing mesenchymal progenitors [23,24]. Although some features of tendon development are autonomous, key stages rely on signals emanating from muscle or cartilage, according to their positioning in the limb [2]. Bone superstructures, which provide anchoring points for tendons to the skeleton, are initiated independently of muscle, but their maintenance and growth depend on cues from both tendon and muscle [2]. Given the distinct gene regulatory networks governing cranial muscle development [25,26], and embryonic origins of connective tissues in the head [7,8], it is unclear if this logic of musculoskeletal integration is conserved in the head and how structures that do not integrate bones, such as EOMs, are established.

All-trans retinoic acid (ATRA), the biologically active metabolite of retinol (vitamin A), is a critical morphogen with widespread roles in craniofacial development [27,28]. ATRA acts as a ligand for nuclear retinoic acid receptors (RARs), which are ligand-dependent transcriptional

regulators that work as heterodimers with retinoid X receptors (RXRs) [28,29]. ATRA is synthetized from retinol through 2 oxidation steps by specific retinol/alcohol dehydrogenases (RDH/ADH) and retinaldehyde dehydrogenases (ALDH1A1, ALDH1A2, and ALDH1A3) [28,29]. ATRA is critical for early eye development in several species [30–32], in which ATRA metabolic enzymes are expressed in the early retina with tight spatiotemporal patterns [33]. As such, the developing eye acts as a signaling center nucleating anterior segment morphogenesis, with paired-like homeodomain transcription factor 2 (PITX2) being the potential major downstream ATRA effector in periocular NCCs [30,34–36]. Whether ATRA is required for morphogenesis of the EOMs and associated connective tissues remains unexplored.

Here, we reassessed the embryological origins of the connective tissues of the periocular region and present the first integrative blueprint for morphogenesis of the EOM functional unit. We provide genetic evidence for the existence of a retinoic acid signaling module that coordinates the emergence of individual EOMs, their tendons, and insertion sites. We show that the action of retinoic acid signaling in muscle patterning is mainly non-cell-autonomous, through its action on the NCC-derived periocular mesenchyme. We propose that the interactions between muscles, tendons, and their attachments are similar to those observed in the limb, yet they exhibit specifc hallmarks that are characteristic of this anatomical location.

## Results

### Genetic fate mapping of mouse periocular tissues

Given the complex anatomical disposition of the EOMs, we first set out to map morphological landmarks and cell relationships during patterning of these muscles. The periocular mesenchyme (POM) is a heterogeneous cell population surrounding the optic cup that gives rise to specialized structures of the anterior segment of the eye and connective tissues associated with the EOMs [37]. With exception of the EOMs and endothelial lining of ocular blood vessels (choroid), all connective tissues of the POM (cartilage, muscle connective tissue, tendons) were reported to be derived from NCCs in zebrafish, chicken, and mouse embryos [38–43]. The 4 recti EOMs originate deep in the orbit, at the level of a fibrous ring called the annulus of Zinn, and insert into the scleral layer of the eye [44]. Because information on EOM tendons and attachment sites is scarce, we used genetic fate mapping to reassess the embryological origins of the tissues interacting with EOMs during their morphogenesis. We simultaneously traced the contribution of NCC (Fig 1A–1B", S1A, and S1B Fig) and mesodermal (Fig 1C–1D", S1C and S1D Fig) derivatives using $Tg:Wnt1^{Cre}$ and $Mesp1^{Cre}$ mice, respectively [45,46], in combination with the $R26_{Tom}$ reporter [74].

As expected, connective tissues at the EOM insertion level were derived from NCCs as assessed by tdTomato expression ($Tg:Wnt1^{Cre};R26^{Tom}$, Fig 1A–1A", S1A Fig). Surprisingly, we found that lineage contributions differed in dorsal sections, where connective tissues at the EOM origin level were derived from the cranial mesoderm ($Mesp1^{Cre};R26^{Tom}$, Fig 1D–1D", S1D Fig). To characterize in more detail the cell populations arising from these derivatives, we used transcription factor 4 (TCF4) as muscle connective tissue marker [47], and a $Scx$ reporter line $(Tg:Scx-GFP+)$ to mark tendons and their early progenitors with green fluorescent protein (GFP) [24,48]. TCF4 was expressed robustly in muscle connective tissue fibroblasts in both NCC- and mesoderm-derived regions (S1E-S1F' Fig). Similarly, $Tg:Scx-GFP$ (Fig 1A–1D", S1A–S1D Fig) strongly labeled the future EOM tendons at the origin and insertion sites residing respectively in mesoderm- and NCC-derived domains. Additionally, $Tg:Scx-GFP$ (Fig 1A1, 1D1, S1G and S1H Fig) and $Scx$ mRNA (S1I and S1J Fig) marked muscle connective tissue fibroblasts that were widely distributed among the muscle masses and overlapped with TCF4 (S1G and S1H Fig), as described in other regions of the early embryo [49,50]. Altogether, these

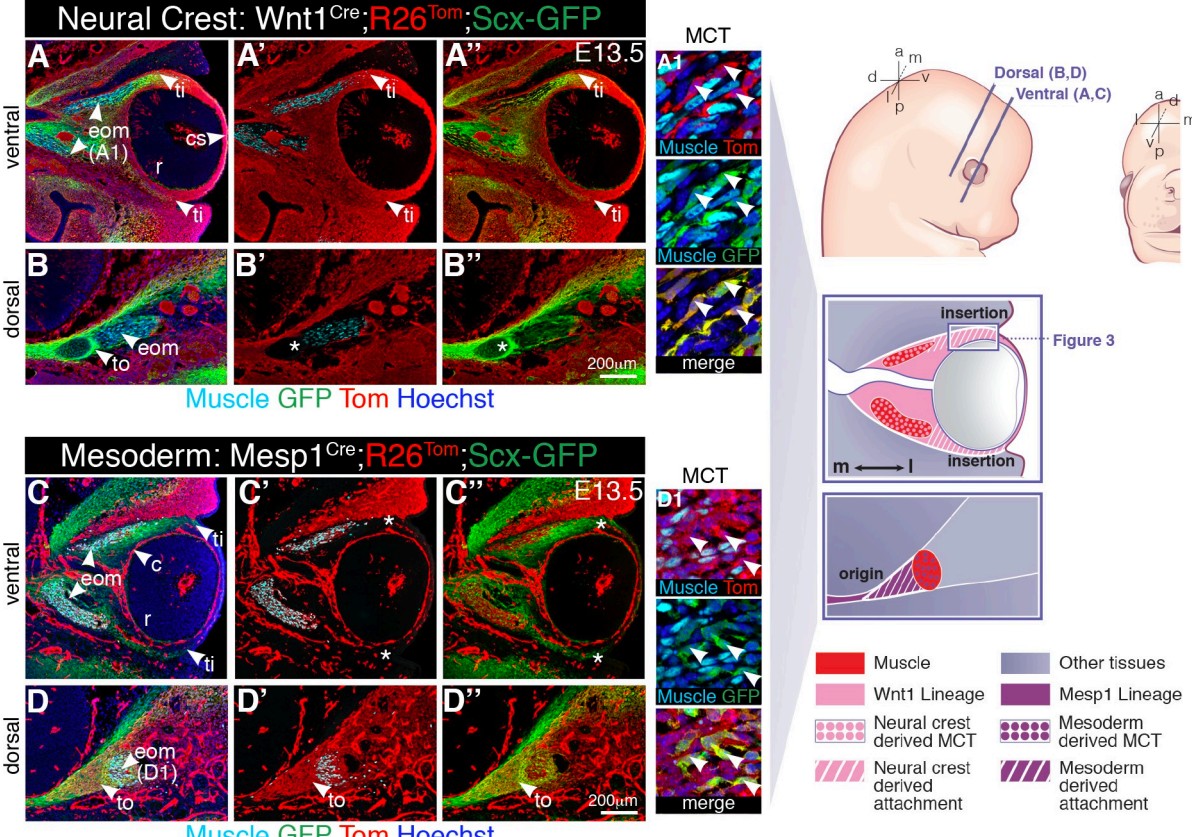

**Fig 1. Lineage contributions to the EOM functional unit. (A-B")** Neural crest (*Tg:Wnt1^Cre;R26^Tom*) and **(C-D")** mesodermal (*Mesp1^Cre;*
*R26^Tom*) lineage contributions to the periocular region of E13.5 embryos, combined with immunostaining for tdTomato (Tom), GFP (*Tg:*
*Scx-GFP* reporter), and muscle (PAX7/MYOD/MYOG, myogenic markers). Coronal sections at ventral (**A-A", C-C"**) and dorsal (**B-B",**
**D-D"**) levels. Asterisks in **B',B"** denote Tom negativity at the tendon origin. Asterisks in **C',C"** denote Tom negativity at the tendon insertion
site. (**A1, D1**) High-magnification views of muscle areas in panels **A** and **D**. (*n* = 3 per condition). a, anterior; c, choroid; cs, corneal stroma;
d, dorsal; E, embryonic day; eom, extraocular muscle; l, lateral; m, medial; MCT, muscle connective tissue; p, posterior; r, retina; ti, tendon
insertion; to, tendon origin; v, ventral.

findings indicate that the EOMs develop in close association with connective tissues of 2 distinct embryonic origins: neural crest laterally (at the EOM insertion) and mesoderm medially (at the EOM origin).

## Development of the EOMs and their insertions overlap spatiotemporally

Patterning of the EOMs and their insertions is understudied because of the difficulty in interpreting a complex 3D tissue arrangement from tissue sections alone. Therefore, we established an imaging pipeline that includes whole-mount immunofluorescence (WMIF) of the periocular region, tissue clearing, confocal microscopy, and reconstructions of the obtained images into 3D objects. To visualize the developing EOMs, we used antibodies against myogenic differentiation 1 (MYOD), myogenin (MYOG), and Desmin as myogenic commitment and differentiation markers and myosin heavy chain (MyHC) to label myofibers (Fig 2A–2C' and S1 Video). At embryonic day (E)11.75, the EOMs were present as a single anlage (Fig 2A and 2A') medial to the eyeball. By E12.5, the EOM anlage split towards the eyeball into submasses corresponding to the future 4 recti, 2 oblique muscles, and the accesory retractor bulbi muscle (Fig 2B and 2B'). Fully individuated muscles were evident by E13.5 (Fig 2C and 2C'). Thus, we

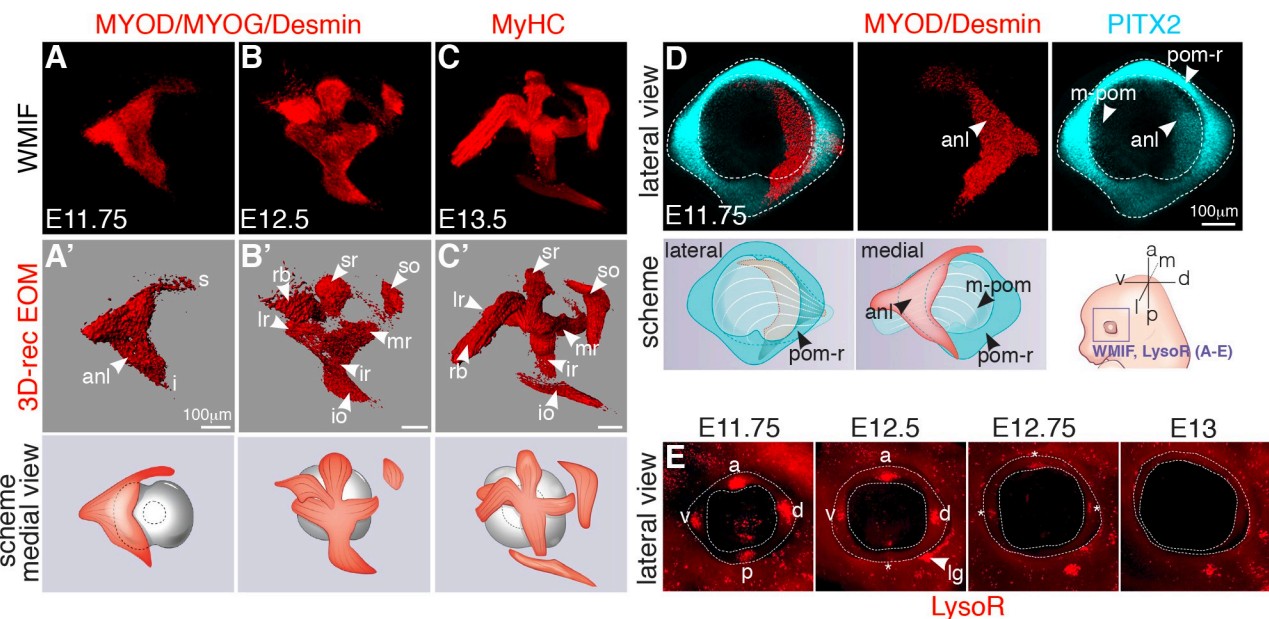

**Fig 2. Developmental time course of EOM development. (A-C)** WMIF for MYOD/MYOG/Desmin (myogenic differentiation markers) **(A, B)** and MyHC (myofibers) **(C)** at the indicated embryonic stages. EOMs were segmented from adjacent head structures and 3D-reconstructed in Imaris (Bitplane). **(A'-C')** EOMs are shown as isosurfaces for clarity of visualization. Medial views as schemes (left eye). **(D)** WMIF for MYOD/Desmin (labeling the EOM anlage, isosurface) and PITX2 (labeling the POM and EOM anlage) on E11.75 embryos (left eye). Lateral and medial views as schemes. **(E)** Whole-mount LysoTracker Red (LysoR) staining of the periocular region at the indicated stages (left eye). The POM is delimited with dashed lines. Asterisks indicate apoptotic foci with reduced intensity from E12.5 onwards. anl, anlage; a, anterior; d, dorsal; 3D-rec, 3D reconstruction; E, embryonic day; i, inferior EOM anlage projection; io, inferior oblique; ir, inferior rectus; l, lateral; lg, lacrimal gland sulcus; lr, lateral rectus; m, medial; m-pom, medial periocular mesenchyme; mr, medial rectus; p, posterior; pom, periocular mesenchyme; pom-r, periocular mesenchyme ring; rb, retractor bulbi; s, superior EOM anlage projection; so, superior oblique; sr, superior rectus; v, ventral; WMIF, whole-mount immunoflurorescence.

conclude that EOM patterning in the mouse occurs by splitting from a single mass of myogenic progenitors that target the eyeball, with the most dramatic morphogenetic changes taking place between E11.75 and E12.5.

The EOMs insert into the sclera, a dense fibrous layer derived from the POM [40,44]. PITX2 is a well-established marker of both the POM and EOM progenitors ([40], S2A–S2C Fig), and foci of cell death in the POM were suggested to label the tendon attachment positions of the 4 recti muscles [51]. To better understand the development of the EOM insertions, we first performed whole-mount immunostaining for the POM marker PITX2 and LysoTracker Red staining to detect programmed cell death on live embryos [52]. WMIF for PITX2 at E11.75 revealed a ring of POM cells expressing high levels of PITX2 that formed a continuum with low-expressing cells extending towards the base of the EOM anlage (Fig 2D and S2 Video). The pattern of LysoTracker Red staining in the POM-ring was highly dynamic, where 4 foci of apoptosis were present at E11.75 at the horizons of the eye but regressed progressively from E12.5 onwards (Fig 2E). To confirm that these foci define tendon attachment positions in the POM, we performed whole-mount imunostaining for myogenic and tendon progenitors on LysoTracker-stained embryos. Surprisingly, we observed that even before muscle splitting initiated, *Scx-GFP*+ condensations bridged the edges of the EOM anlage and the 4 LysoTracker Red+ foci in the POM (E11.75, S3 Video), presaging the attachment sites of the future 4 recti muscles (E12.5, S4 Video).

We next wanted to understand the relationship between foci of apoptosis in the POM and the establishment of the tendon insertion sites per se. In the developing limb and jaw, bone superstructures or ridges, generated by a unique set of progenitors that co-express *Scx* and

SRY-box containing gene 9 (SOX9), provide a stable anchoring point for muscles via tendons [53–56]. Although EOMs insert into a non-bone NCC-derived structure, early markers of pre-committed cartilage, such as SOX9, are expressed in the POM [57]. To examine the time course of development of EOM insertions in greater detail, we immunostained *Tg:Scx-GFP* and *Tg*:*Wnt1^{Cre}*;*R26^{Tom}*;*Scx-GFP* coronal (Fig 3A–3J" and S3A–S3F' Fig) and transverse (S3G–S3H" Fig) sections for PITX2 and SOX9. Between E11.5 and E13.5, PITX2 marked all cells of the lateral-most NCC-derived POM between the surface ectoderm and retina (Fig 3B and 3F, and S2A–S2B" Fig), which corresponds to the POM-ring observed in 3D views (Fig 2D). In this region, *Scx-GFP* expression was initially detected in a salt-and-pepper pattern (Fig 3C and S3E Fig) but became progressively limited to the forming tendon tips (Fig 3G and S3F Fig). SOX9 expression overlapped with that of PITX2 (Fig 3D, 3H, S3E' and S3F' Fig) but became more restricted to the insertion site by E13.5 and with a pattern complementary to *Scx-GFP*+ (Fig 3G and S3F Fig). Notably, *Scx-GFP*+ SOX9+ cells could be detected between E11.5 and E13.5 at the interface between mutually exclusive *Scx-GFP*+ and SOX9+ cells (S3A–S3C" Fig), resembling what was observed during tendon-to-bone attachment formation in other regions in the embryo [53–56].

At the putative insertion site, the tdTomato staining in the NCC-derived lateral POM initially appeared as punctate (Fig 3A), and reminiscent of the apoptotic domains observed in 3D views (Fig 2E). LysoTracker Red and TUNEL staining confirmed cell death of SOX9+ PITX2+ POM cells (Fig 3I–3J", S3E, S3E' and S3G–S3H" Fig) that were at a higher density than the more medial POM cells (Fig 3K, S1 Data). Given that LysoTracker Red+ cells could not longer be seen at E13.5 (S3F and S3F' Fig), these data suggest that at the insertion sites of the recti muscles in the POM, foci of apoptosis mark the places where cell compaction and refinement of the SOX9 expression pattern will take place.

Major POM remodeling events could be detected by E14.5. EOM tendons co-expressed *Scx-GFP*, Tenascin, and PITX2, but surprisingly, SOX9 expression became restricted to the thin scleral layer and retinal pigmented epithelium (RPE) (Fig 3L–N' and S3D–S3D" Fig). Altogether, these results show that development of the EOMs, their tendons and insertion sites overlap spatiotemporally and thus, might be regulated in a coordinated manner. Moreover, similarly to other locations in the body, *Scx* and SOX9 show dynamic expression patterns at the insertion site, but additional specifc hallmarks, notably the presence of cell compaction and apoptotic foci, seem to be characteristic of this anatomical location.

## Abnormal EOM morphogenesis in mutants with ocular malformations

Having assessed how morphogenesis of EOMs and their insertion sites is coordinated, we set out to investigate the role of the target organ, the eyeball, on the establishment of the EOM functional unit. To this end, we performed micro-computed tomography (micro-CT) scans in mouse mutants with a spectrum of ocular perturbations. First, we examined *small eye* (Sey) *Pax6* (Paired box 6) mutant embryos (*Pax6^{Sey/Sey}*), in which eye development is arrested at the optic vesicle stage [58–60]. EOM patterning in *Pax6^{Sey/+}* embryos proceeded normally (S4A and S4B Fig), whereas in the *Pax6^{Sey/Sey}* mutant, the EOMs appeared as a single mass on top of a rudimentary optic vesicle (S4C Fig). As *Pax6* is expressed in the optic vesicle and overlying ectoderm that forms the lens and cornea [58], but not in EOMs, these observations suggest a non-cell-autonomous role in EOM patterning. Similarly, in LIM homeobox 2 (Lhx2) mutant embryos (*Tg:Lhx2^{Cre}*;*Lhx2^{fl/fl}*) embryos, in which inactivation of the *Lhx2* gene in eye committed progenitor cells leads to a degeneration of the optic vesicle at E11.5 [61], EOM patterning was severely affected, and few EOM submasses were observed (S4D–S4E' Fig). Finally, in cyclopic embryos resulting from inversion of Sonic hedgehog (Shh) regulatory regions, EOMs

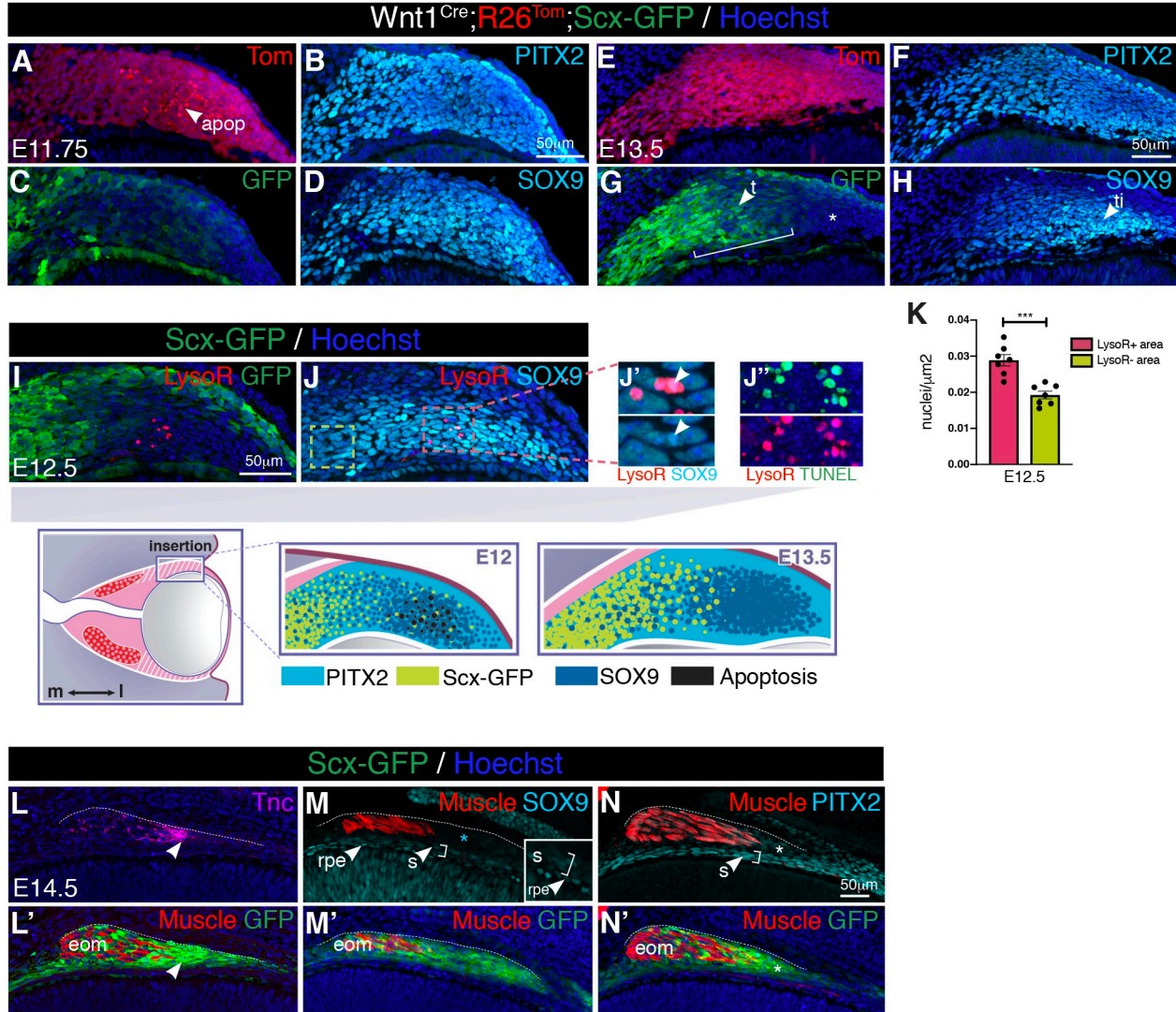

**Fig 3. Developmental time course of EOM insertions in the POM. (A-D)** Immunostaining of lateral POM in coronal sections of E11.75 *Tg*: *Wnt1^Cre^;R26^Tom^;Scx-GFP* embryos for the indicated markers. **B**, Immunostaining of section adjacent to the one shown in (**A, C, D**; channels split for clarity). **(E-H)** Immunostaining of lateral POM in coronal sections of E13.5 *Tg:Wnt1^Cre^;R26^Tom^;Scx-GFP* embryos for the indicated markers. **G**, Immunostaining of section adjacent to the one shown in (**E, F, H**; channels split for clarity). Bracket show overlap between *Scx-GFP* and SOX9 expression domains (high-magnification views in S3C" Fig). The asterisk in **G** points to *Scx-GFP*-negative SOX9+ lateral-most POM at the insertion site. **(I,J)** Immunostaining of lateral POM in coronal sections of E12.5 *Tg:Scx-GFP* embryos pretreated with LysoR. Arrowheads in **J'** point to LysoR + SOX9+ cells. **(J")** TUNEL and LysoR staining of a section adjacent to the one shown in (**I,J**). **(K)** Quantification of the total number of SOX9+ cells per square-micrometer in LysoR+ regions (red square, **J**) and more medial LysoR-negative regions (yellow square, **J**). Mann–Whitney test. Cell density was 33% higher in the LysoR+ area compared with the more medial POM. See S1 Data for individual values. **(L-N')** Immunostaining on coronal sections of E14.5 *Tg:Scx-GFP* embryos. **(L,L')** Tnc and *Scx-GFP* co-localize in tendons at level of insertion (arrowhead). **(M,M')** SOX9 expression in the POM is greatly reduced at the insertion site and no longer overlaps with *Scx-GFP* (asterisk, cyan). Low levels of SOX9 expression in the RPE and sclera. **(N,N')** PITX2 remains expressed in the POM at the insertion site overlapping with *Scx-GFP* (asterisk) and in the sclera. MyHC (**L'**) and SMA (**M-N'**) were used to label EOM muscle. Dashes in **L-N** were drawn according to GFP labeling in L'-N'. Images in **A-N'** correspond to insertion site of superior rectus muscle in the POM as shown in the scheme. (*n* = 3 per condition). a, anterior; apop, apoptosis spots; d, dorsal; E, embryonic day; eom, extraocular muscle; l, lateral; LysoR, LysoTracker Red; m, medial; p, posterior; POM, periocular mesenchyme; rpe, retinal pigmented epithelium; s, sclera; t, tendon; ti, tendon insertion; v, ventral.

underwent splitting and projected towards the centrally located ectopic eye, although with an abnormal 3D arrangement as reported for human cyclopia conditions (S4F–S4F" Fig) [7]. Together with previous studies in which surgical removal of the eye at specific timepoints of development results in smaller EOMs [9,62], our observations point to the eye as a critical organizer of EOM patterning.

## Muscle patterning depends on retinoic acid signaling of neural origin

To study the role of target organ derived cues in EOM patterning, we investigated the role of retinoic acid signaling, which plays multiple paracrine roles during embryonic eye development [33]. As ALDH1A1-3 are rate-limiting enzymes in the production of ATRA [63], we characterized their expression in the periocular region at the time of EOM patterning. Between E10.5 and E12.5, *Aldh1a1* was expressed strongly in the dorsal retina and lens, *Aldh1a2* was expressed in the temporal mesenchyme, and *Aldh1a3* was most strongly expressed in the ventral retina and RPE (S5A–S5A" Fig) [34,36]. At the protein level, ALDH1A3 was detected on tissue sections in the surface ectoderm, presumptive corneal epithelium, retina, and RPE (Fig 4A). Interestingly, ALDH1A3 was also expressed in the optic stalk between E10.5 and E12.5 and thus centrally positioned with respect to EOM development (Fig 4A). To target the retinoic signaling pathway (Fig 4B), we used *Aldh1a3*[-/-] [64] and *Rdh10*[-/-] [65] mutants. We also administered the BMS493 inhibitor (pan-RAR inverse agonist) to pregnant females every 10–12 hours between E10.5 and E11.75, i.e., preceeding the initiation of muscle splitting (Fig 2A–2C), and once NCC migration to the POM was finalized [66,67]. Micro-CT analysis showed that *Aldh1a3*[-/-] and BMS493-treated embryos displayed eye ventralization and a shortened optic nerve when compared with controls (S5B Fig), but they retained the overall organization of the nasal capsule and orbit (S5C Fig). Ventral and lateral views of the 3D-reconstructed EOMs (Fig 4C–4E), showed that *Aldh1a3*[-/-] and BMS493-treated embryos lacked the standard 3D arrangement of 4 recti and 2 oblique muscles observed at E13.5 in control embryos. Nevertheless, in all cases, EOMs originated medially from the hypochiasmatic cartilages of the presphenoid bone, indicating that the overall orientation of the EOMs was preserved (Fig 4D"'). Given that the EOMs are more affected at their insertion than their origin level upon ATRA deficiency, this finding suggests that EOM patterning is, in part, modular.

To analyze EOM and tendon patterning with higher resolution, we performed whole-mount immunostainings for differentiated myofibers and tendon with MyHC and Tnc (Tenascin) antibodies (Fig 4F–4H'). On medial and lateral views of 3D-reconstructed EOMs, only the retractor bulbi and superior rectus could be clearly identified among the non-segregated muscle fibers in *Aldh1a3*[-/-] embryos (Fig 4G and 4G'). As expected from global invalidation of retinoic acid signaling, EOM perturbation was more severe in BMS493-treated embryos (Fig 4H and 4H'). The superior oblique was absent or continuous with the anterior part of the anlage, and the medial portion of the retractor bulbi was thicker and less clearly isolated from the rest of the anlage (Fig 4H). In both conditions of ATRA deficiency, Tenascin and *Scx-GFP+* cells were present at the tips of the individual, though mispatterned, muscles at this stage (Fig 4G', 4H', and S5 Video). Analysis of *Aldh1a3*[-/-] and BMS493-treated embryos revealed, on average, a 26% reduction in the EOM volume compared with controls (Fig 4I, S2 Data). However, MyHC+ myofibers were present in *Aldh1a3*[-/-] and BMS493-treated embryos (Fig 4G and 4H), suggesting that EOM differentiation was not overtly affected in these conditions. Instead, these observations suggest that EOM fiber alignment and segregation of the muscle masses are dependent on retinoic acid signaling.

As dose and temporal control are critical in the context of retinoic acid signaling [27,32,68], we performed other BMS493 injection regimes between E10.5 and E12.5 (S1 Table). EOM

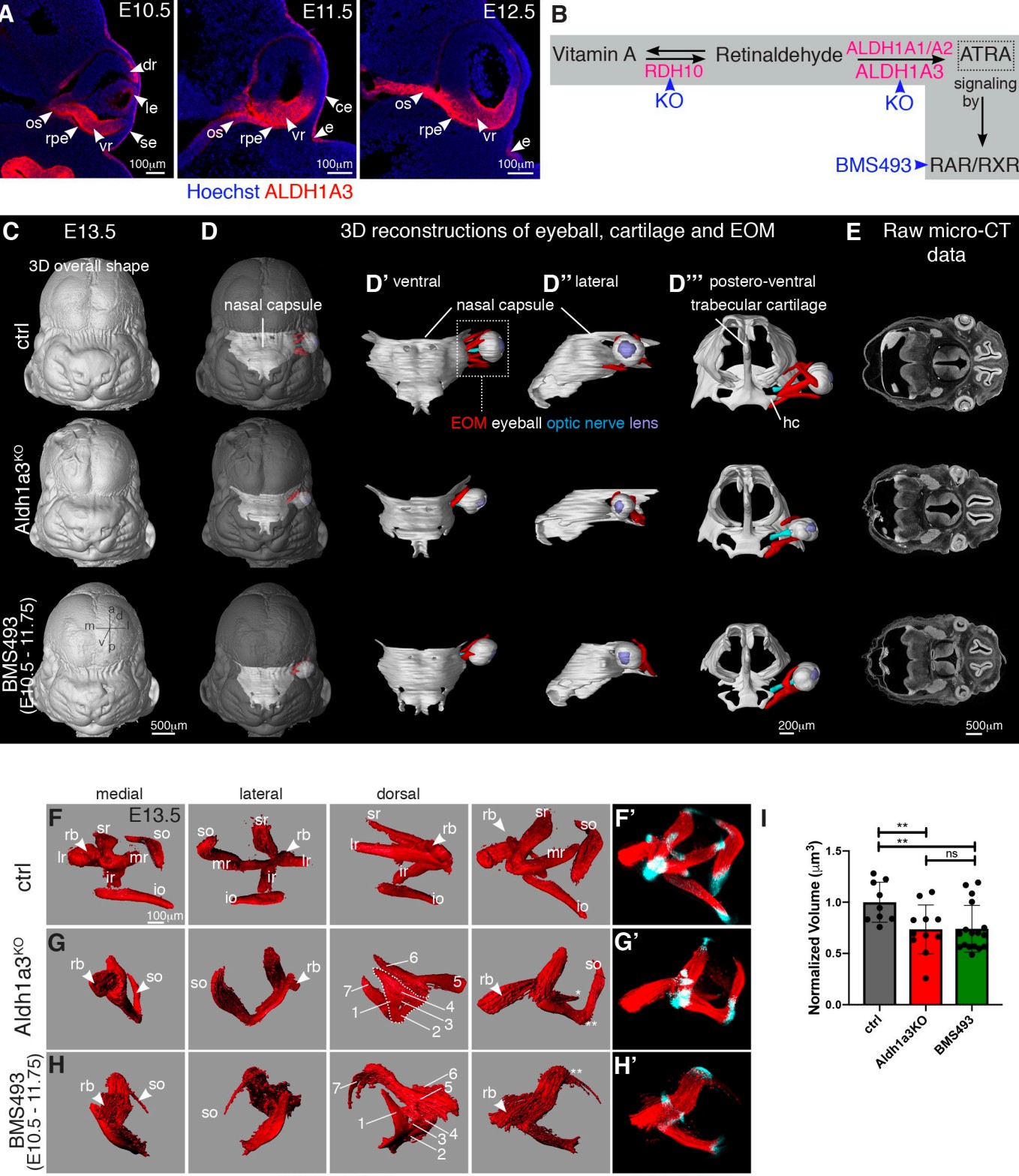

**Fig 4. Extraocular muscle morphogenesis is dependent on ATRA. (A)** Immunostaining for ALDH1A3 on coronal sections of E10.5, E11.5, and E12.5 control embryos (*n* = 3). **(B)** Scheme of retinoic acid signaling pathway with key enzymes for oxidation of retinol (Vitamin A) and retinaldehyde (pink) and mutants/inhibitors used in this study (blue). **(C-D''')** Micro-CT-based 3D-reconstruction of chondrogenic mesenchymal condensations of nasal capsule, trabecular

cartilage, EOM, eyeball, optic nerve, and lens in E13.5 control, *Aldh1a3*$^{KO}$ and BMS493-treated embryos. EOM visualization in context of whole head (**D**), nasal capsule (**D'-D"**), trabecular cartilage (**D'''**). (**E**) Raw micro-CT data (*n* = 2). (**F-H**) 3D-reconstructions of WMIF for MyHC of E13.5 control (**F**), *Aldh1a3*$^{KO}$ (**G**) and BMS493-treated embryos (**H**) (*n* > 9). EOMs were segmented from adjacent head structures and 3D-reconstructed in Imaris (Bitplane). EOMs are shown as isosurfaces for clarity of visualization. (1–7) denote non-segregated muscle masses with differential fiber orientation (see also S5D Fig). Raw immunostaining data for MyHC (myofibers) and Tnc (tendon) are shown in panels **F'-H'**. Double asterisk in **G,H** indicate fused muscle masses. Asterisk in **G** indicate misoriented medial rectus. (**I**) Relative EOM volume (compared with control) of WMIF in **F-H**. Each dot represents an individual embryo (*n* > 9). Mann–Whitney test. See S2 Data for individual values. a, anterior; ATRA, all-trans retinoic acid; ce, presumptive corneal epithelium; ctrl, control; d, dorsal; dr, dorsal retina; 3D-rec, 3D-reconstruction; e, eyelid groove; E, embryonic day; EOM, extraocular muscle; hc, hypochiasmatic cartilage; io, inferior oblique; ir, inferior rectus; l, lateral; le, lens; lr, lateral rectus; m, medial; mr, medial rectus; micro-CT, micro computed tomography; os, optic stalk; p, posterior; rb, retractor bulbi; rpe, retinal pigmented epithelium; RAR, retinoic acid receptor; RXR, retinoid X receptor; se, surface ectoderm; so, superior oblique; sr, superior rectus; v, ventral; vr, ventral retina.

patterning phenotypes categorized as strong or severe at E13.5 were only obtained when an E10.75 time point of injection (8 PM of day E10.5) was included in the regime (experiment type I-III, S1 Table, and S5D Fig). Surprisingly, even a single injection at this time point resulted in strong phenotypes (experiment type IV, S1 Table, and S5D Fig), whereas exclusion of this time point (experiment type V, S1 Table, and S5D Fig) resulted in only mild phenotypes at best. Therefore, we identified an early and restricted temporal window in which ATRA activity, prior to any sign of muscle splitting, impacts correct muscle patterning 36–48 hours later. Given the critical action of RDH10 in generating retinaldehyde, the intermediate metabolite in the biosynthesis of ATRA, we examined EOM development in *Rdh10* mutant embryos. *Rdh10* is normally expressed in the optic vesicle and RPE [69], and in *Rdh10*$^{-/-}$ embryos, eyes develop intracranially, close to the diencephalon with a very short ventral retina [65]. WMIF analysis of *Rdh10* mutant EOMs revealed that the anlage was specified but lacked any sign of segmentation, in agreement with an upstream role in ATRA synthesis (S5E and S5F Fig). Taken together, our data show that retinoic acid signaling is essential for the correct myofiber alignment and segregation of EOM masses in a dose- and time dependent manner.

## ATRA-responsive cells drive muscle patterning in the periocular mesenchyme

In all body regions, muscle connective tissue plays a central role in muscle patterning [22]. Moreover, this process appears to be tightly coupled to development of tendons and tendon attachment sites [15,70–72]. To determine whether the role of retinoic acid signaling in EOM patterning is direct in myogenic cells or indirect, through action in adjacent connective tissues, we tracked cells that are responsive to ATRA using a novel retinoic acid transgenic Cre reporter line [73]. This transgenic line comprises 3 retinoic acid response elements *(RARE)* from the *Rarb* gene fused to the *Hspa1b* minimal promoter driving expression of a tamoxifen-inducible *Cre-ER*$_{T2}$ recombinase (*Tg:RARE-Hspa1b-Cre/ER*$^{T2}$, designated as *Tg:RARE-CreERT2* for simplicity). By crossing this line with the *R26*$^{mTmG}$ or *R26*$^{Tom}$ reporter mice [74,75] (Fig 5A), we permanently labeled ATRA-responsive cells and their descendants with membrane-tagged GFP or tdTomato. Different tamoxifen regimes showed that a greater number of responsive cells were present in the periocular region when tamoxifen was administered between E9.75–E10.5 (S6A–S6C Fig). This finding is in agreement with BMS493 treatments identifying E10.75 as a critical time point for the action of ATRA (S1 Table and S5D Fig) and the fact that maximal recombination efficiency can be achieved between 12–24 hours upon tamoxifen induction [76,77].

To assess whether myogenic cells and adjacent POM cells respond to retinoic acid signaling, we microdissected the periocular region of *Tg:RARE-CreERT2;R26*$^{mTmG}$ embryos and subjected it to mild digestion in bulk. Cells were allowed to attach to culture dishes, immunostained, and scored for co-expression of GFP and myogenic markers (MYOD, MYOG). Notably, the great majority of reporter positive-responsive cells were not myogenic (Fig 5B and 5C,

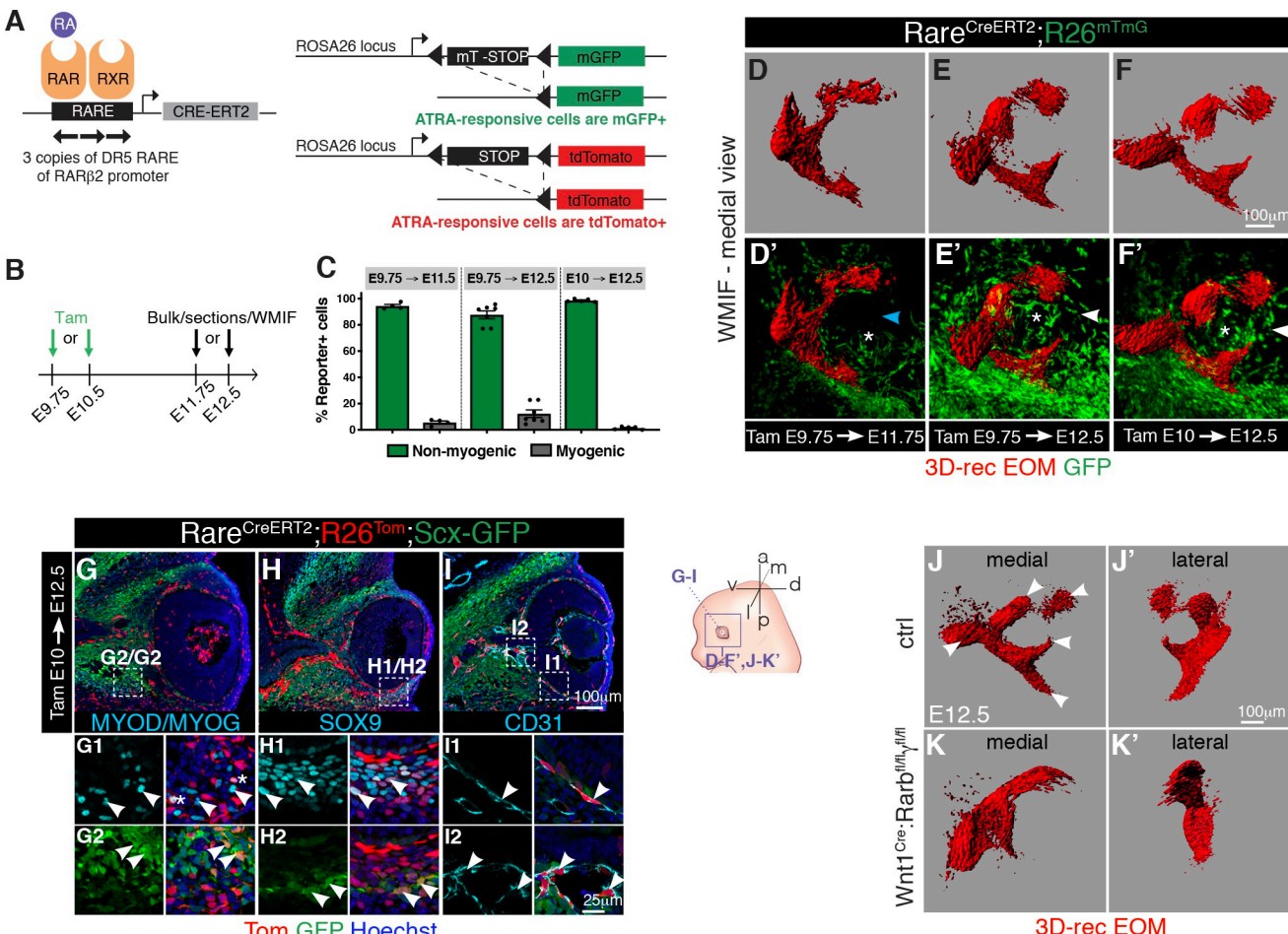

**Fig 5. Periocular connective tissues are responsive to retinoic acid signaling. (A)** Scheme of mouse alleles used. **(B)** Strategy used to determine cell types responsive to retinoic acid signaling in *Tg:RARE-CreERT2;R26^mTmG* embryos. Tamoxifen was injected to pregnant females at E9.75 or E10.5 and analysis performed (bulk, sections or WMIF) at E11.75 or E12.5. **(C)** The percentage of recombined (GFP+) cells within the non-myogenic or myogenic populations (PAX7+, MYOD+, MYOG+) was assessed by immunostaining on bulk cell preparations of the periocular region of *Tg:RARE-CreERT2; R26^mTmG* embryos following different tamoxifen treatments. Each dot represents an individual embryo (*n* > 4 embryos/condition). See S3 Data for individual values. **(D-F')** WMIF of *Tg:RARE-CreERT2;R26^mTmG* embryos for SMA (differentiated muscle) and GFP (ATRA-responsive cells) at the indicated embryonic stages. The number of reporter positive cells at the place where the developing medial rectus muscle will project increases from E11.75 (blue arrowhead) to E12 (white arrowheads). Asterisks mark the optic nerve (*n* = 3). **(G-I)** Coronal sections of E12.5 *Tg:RARE-CreERT2;R26^Tom;Scx-GFP* embryos immunostained for GFP, Tom (ATRA-responsive cells), MYOD/MYOG (muscle), and CD31 (endothelial cells). Higher-magnification views as insets. Asterisks in G1 indicate sporadic labeling in myogenic cells (*n* = 3). **(J-K')** WMIF for MyHC (myofibers) of E12.5 control (*Tg:Wnt1Cre;Rarb^fl/+; Rarg^fl/+*) **(J,J')** and mutant (*Tg:Wnt1^Cre;Rarb^fl/fl;Rarg^fl/fl*) **(K,K')** EOM. Medial and lateral views are shown. Note absence of splitting in mutant EOM. Arrowheads indicate split EOM in controls (*n* = 2 per genotype). In **D-F'** and **J-K'** EOMs were segmented from adjacent head structures and 3D-reconstructed in Imaris (Bitplane). EOMs are shown as isosurfaces for clarity of visualization. a, anterior; ATRA or RA, all trans retinoic acid; ctrl, control; d, dorsal; 3D-rec, 3D-reconstruction; E, embryonic day; EOM, extraocular muscle; l, lateral; m, medial; mGFP, membrane-tagged GFP; p, posterior; RAR, retinoic acid receptor; RXR, retinoid X receptor; SMA, alpha smooth muscle actin; Tam, Tamoxifen; v, ventral; WMIF, whole-mount immunofluorescence.

S6D Fig and S3 Data). We next performed immunostaining in whole-mount to assess the spatial distribution of ATRA-responsive cells in periocular connective tissues (Fig 5D–5F'). This analysis revealed that the mesenchyme around the optic nerve in close proximity to the developing EOMs was positive for the reporter. Interestingly, the number of POM GFP+ cells increased from E11.75 to E12 (Fig 5D'–5F'), which is the temporal window corresponding to EOM splitting (Fig 2A and 2B). Inactivation of downstream retinoic acid signaling with BMS493, prior and subsequent to the induction of *Tg:RARE-CreERT2;R26^mTmG* mice with

tamoxifen (S6E Fig), greatly reduced the responsiveness of the reporter in the POM compared with non-treated controls (S6F–S6I Fig, S6 and S7 Videos).

As with experiments on isolated cells, the great majority of tdTomato positive-responsive cells on tissue sections of *Tg:RARE-CreERT2;R26^{Tom}* embryos were not myogenic at ventral (Fig 5G) and dorsal levels (S6J Fig). tdTomato positive-responsive cells co-expressed *Scx-GFP* (Fig 5G and 5H), Sox9 (Fig 5H) or the endothelial cell marker CD31 along the choroid (vascular layer of the eye) and ciliary arteries around the optic nerve (Fig 5I). Together, these results are in agreement with previous studies [34–36] showing that despite a sophisticated pattern of ATRA metabolism in the developing retina, retinoic acid signaling exerts its action mostly non-cell autonomously. We show that ATRA targets various connective tissue types of the POM within a temporal window that is crucial for EOM patterning.

As ATRA-responsive cells were found in NCC- (ventral POM) and mesoderm-derived cell compartments (dorsal POM, choroid, and a fraction of myogenic progenitors), we performed more selective perturbations of retinoic acid signaling with available genetic tools. Expression of a dominant negative nuclear retinoid receptor isoform in myogenic cells (*Myod^{iCre}*; *R26^{RAR403}*, [78,79]) did not result in noticeable EOM patterning defects (S6K–S6L' Fig). We then inactivated the RARβ and RARγ receptors in the NCC-derived POM, which respond to ATRA synthesized by the retina [80,81], using the *Tg:Wnt1^{Cre}* driver (*Tg:Wnt1^{Cre};Rarb^{fl/fl}*; *Rarg^{fl/fl}*, [35]). WMIF for muscle in *Tg:Wnt1^{Cre};Rarb^{fl/fl};Rarg^{fl/fl}* embryos showed absence of muscle splitting (Fig 5J–5K'), and this was as severe as that observed in fully ATRA-deficient *Rdh10^{-/-}* embryos (S5F Fig). Altogether, these data suggest that synthesis of ATRA from neural derivatives (retina, optic nerve, RPE) is crucial for EOM patterning, through its action on NCC-derived periocular connective tissues at earlier stages.

## Defective organization of EOM insertions in embryos deficient in retinoic acid signaling

Having adressed a major role of ATRA in NCC-derived cells of the POM for EOM patterning, we set out to assess which specific connective tissue subpopulations, including the EOM insertions in the POM, were affected in *Aldh1a3^{-/-}* and BMS493-treated embryos. First, we examined the distribution of Collagen XII (Fig 6A–6C'), a marker of the sclera and corneal stroma [82], on tissue sections at E12. We observed a marked reduction of Collagen XII (Col XII) at the level of the medial-POM that gives rise to the sclera, upon ATRA deficiency (Fig 6B–6C'). In addition, although PITX2 was expressed continuously in the entire POM and corneal stroma in controls (Fig 6D and 6D'), expression in the medial-POM was selectively lost upon ATRA deficiency (Fig 6E–6F'). This observation is in agreement with *Pitx2* being a retinoic acid signaling target in the NCC-derived POM [34–36] and the sclera being absent at fetal stages in *Rarb/Rarg*-NCC mutants [35]. Interestingly, POM organization was unaffected in *Myf5^{nlacZ/nlacZ}* mutants (Fig 6G and 6G'), in which the initial EOM anlage forms but myogenesis is aborted from E11.5 [83].

To understand whether the abnormal distribution of POM markers translates into EOM insertion defects and how EOM development and insertion sites are coordinated, we examined the organization of the POM-ring in control, *Aldh1a3^{-/-}*, and BMS493-treated embryos. We systematically compared E11.75 and E12 embryos to visualize the organization of these tissues prior to and during muscle splitting (Fig 6H–6P and S7A–S7N Fig). Control embryos showed the presence of overlapping SOX9 and PITX2 POM-ring domains, together with 4 foci of apoptosis labeling the prospective tendon attachment points of the 4 recti muscles (Fig 6H–6J, S7A–S7C', S7E and S7F Fig). As expected, ATRA deficiency perturbed these patterns in a dose-dependent manner. At the level of the POM-ring, *Aldh1a3^{-/-}* embryo heads displayed

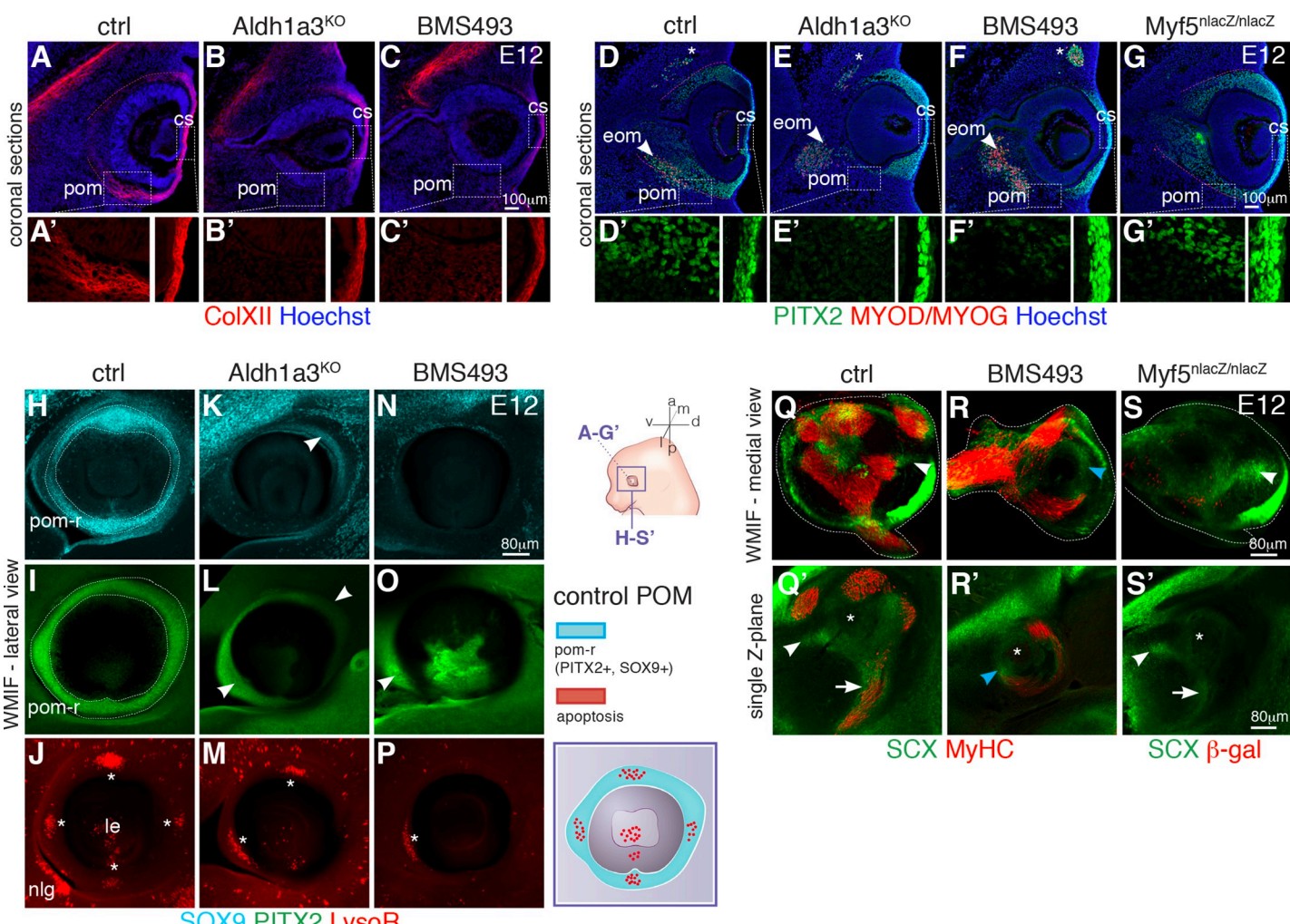

**Fig 6. Altered EOM insertions in the POM upon ATRA deficiency. (A–C')** Immunostaining for ColXII on E12 coronal sections. ColXII expression is lost in the medial POM (future sclera) but not in the corneal stroma of *Aldh1a3$^{KO}$* (**B,B'**) and BMS493-treated embryos (**C,C'**). Higher magnification of ColXII staining as insets in (**A'–C'**). **(D-G)** Immunostaining for PITX2 (muscle progenitors, POM) and MYOD/MYOG (myogenic cells) on E12 coronal sections. Asterisks indicate superior oblique muscle. Red dashed lines delimitate the POM in controls (**D**) and *Myf5$^{nlacZ}$* (**G**) embryos. Higher magnification of PITX2 staining as insets in (**D'-G'**). Note absence of PITX2 expression in medial-POM of *Aldh1a3$^{KO}$* (**E,E'**) and BMS493-treated embryos (**F,F'**). **(H-P)** WMIF for SOX9 and PITX2 of E12 control (**H-J**), *Aldh1a3$^{KO}$* (**K-M**) and BMS493-treated (**N-P**) embryos preincubated with LysoR (left eyes). *Aldh1a3$^{KO}$* and BMS493 embryos do not show a complete POM-ring as controls (dashed lines in **H,I**). Arrowheads mark remaining expression of SOX9/PITX2 in the POM of *Aldh1a3$^{KO}$* (**K,L**) or inhibitor-treated (**O**) embryos. Asterisks in (**J, M, P**) mark apoptosis spots in POM. (**Q-R**) WMIF for MyHC (myofibers) and GFP on E12 control (**Q**) and BMS493-treated *Tg:Scx-GFP* embryos (**R**). WMIF for β-gal (myogenic progenitors) and GFP on E12 *Tg:Scx-GFP;Myf5$^{nlacZ/nlacZ}$* E12 embryos (**S**) (left eyes). The periocular area was segmented from adjacent structures for ease on visualization (dashed lines). White arrowheads in **Q** and **S** highlight correct position of tendon condensations for medial rectus muscle. Blue arrowhead in R marks a diffuse *Scx-GFP+* pattern at the position of the mispatterned medial rectus muscle. (**Q'-S'**) Single Z-section of the segmented volume. White arrowheads in **Q',S'** show tendon tips and white arrows correct *Scx-GFP+* connective tissue pattern along with muscle (**Q'**) or prospective muscle masses (**S'**). Blue arrowhead in R' marks a *Scx-GFP+* condensation at the muscle tip. Asterisks mark position of the optic nerve. (*n* = 3 per condition). a, anterior; ATRA, all-trans retinoic acid; cs, corneal stroma; ctrl, control; d, dorsal; eom, extraocular muscles; l, lateral; le, lens; LysoR, LysoTracker red; m, medial; nlg, nasolacrimal gland; p, posterior; pom, periocular mesenchyme; pom-r, periocular mesenchyme ring; v, ventral; WMIF, whole-mount immunofluorescence.

narrower SOX9 and PITX2 expression domains than controls (Fig 6K–6L, S7G and S7H Fig). In BMS493-treated embryos, expression of these genes in the POM-ring (Fig 6N, 6O, S7D, S7D', S7I and S7J Fig) was further reduced to a minority of cells. Accordingly, foci of apoptosis were lost or disorganized in both cases of ATRA deficiency with a single LysoTracker Red spot almost invariably present in the ventral POM in both conditions (Fig 6M, 6P, S7H and S7J Fig). In medial views of reconstructed 3D volumes, PITX2 showed continuous expression

from the POM-ring to the medial-POM in control embryos (Fig 2D, S7K Fig and S2 Video), whereas we noticed loss of PITX2 expression in both cases of ATRA deficiency (S7M and S7N Fig, blue arrowheads) as observed on tissue sections (Fig 6E' and 6F'). Consistent with these results, the *Tg:RARE-CreERT2;R26$^{mTmG}$* reporter showed a marked reduction of ATRA-responsive cells in the medial POM and POM-ring of *Aldh1a3$^{-/-}$* and BMS493-treated embryos when compared with controls (S7O-S7Q' Fig and S8–S10 Videos). Thus, as seen in other body regions, where development of muscles and bone superstructures into which they insert are tightly coordinated [70], the development of the EOMs is tightly coupled to development of their insertion sites in the soft tissues of the POM.

As tendons were present at muscle tips in E13.5 *Aldh1a3$^{-/-}$* and BMS493-treated embryos but located ectopically with respect to a wild-type configuration (Fig 4F'–4J'), we suspected that the initial 3D organization of *Scx-GFP*+ tendon and connective tissue progenitors might also be affected. We performed WMIF for differentiated muscle (MyHC) and tendon progenitors (*Tg:Scx-GFP*) and segmented the reconstructed 3D volume. In control embryos at E12.5, *Scx-GFP*+ condensations projected from the tips of the future 4 recti muscles toward the POM-ring (Fig 6Q and S11 Video), facing the 4 foci of apoptosis (Fig 6J and S4 Video). Moreover, as observed in tissue sections (Fig 1A1, 1D1, and S1G–S1H' Fig), *Scx-GFP*+ and TCF4 + cells were also present in POM connective tissues presaging the final locales of the recti muscles (Fig 6Q' and S7R Fig). In BMS493-treated embryos at E12, *Scx-GFP* and TCF4 expression were less organized and diffuse (Fig 6R and 6R', and S7S Fig), yet *Scx-GFP*+ condensations started to form at the tips of ectopic muscle masses (Fig 6R' and S12 Video). Altogether, these results show that ATRA is required for the proper specification/organization of NCC-derived POM connective tissues, which includes the muscle connective tissue, tendons, and their insertion sites.

In stark contrast with what we observed upon ATRA deficiency, *Scx-GFP* and TCF4 prepatterning in POM connective tissues was preserved in muscle-less *Myf5$^{nlacZ/nlacZ}$* mutants (Fig 6S and 6S', and S7T Fig). Moreover, *Scx-GFP*+ tendon condensations formed at the tips of prospective (albeit absent) recti muscles (Fig 6S and 6S'). This situation is similar to the especification of individual tendons on muscle-less limbs [20,84] and branchial arches [41]. However, at later stages, tendons did not continue their maturation in the absence of EOMs (S7V and S7V' Fig, S13 and S14 Videos), reinforcing the notion that, as in most other anatomical locations [2], tendon differentiation and maintenance ultimately rely on muscle–tendon interactions.

In summary, ATRA produced by the developing eye influences the organization of the entire NCC-derived POM, allowing integration of these tissues into a musculoskeletal funtional unit (S8 Fig).

## Discussion

The striking differences in embryological origins, function, and susceptibility to disease among cranial and somite-derived muscles [26] provides impetus to study in detail the morphogenesis of different muscle groups and their integration as part of the musculoskeletal system. Here, we focused on EOMs, an evolutionary conserved group of muscles that are precisely engineered for fine displacement of the eyeball and thus, crucial for visual acuity. Using genetic and 3D imaging approaches, we analyzed EOM development from their emergence as a unique anlage to the establishment of a fully formed functional unit with insertions in the sclera, which sets it apart from classical muscle-to-bone insertions studied to date. We identified a spatiotemporal window in which retinoic acid signaling from the target organ is required for patterning of NCC-derived soft tissues of the POM. These findings provide insights into the deployment of site-specific programs for the establishment of anatomically

distinct muscle functional units, with precise definition of muscle shapes and topographical wiring of their tendon attachments.

## Dual origin of EOM insertions

The connective tissues of the rostral cranium (bone, cartilage, tendon, and muscle connective tissue) were reported to be derived from the neural crest [3,7,8]. However, the hypochiasmatic cartilages, from which the recti muscles originate in mammals, are an exception to this rule as they are mesoderm derived [85]. Here, we found that at this location, tendon and muscle connective tissue are also of mesodermal origin. As such, EOMs originate in mesoderm-derived bone and insert in NCC-derived fibrous tissue of the sclera with muscle connective tissue following the embryonic origins of the respective attachments, as observed in other anatomical locations [86]. Taken together, our observations redefine a novel boundary for NCC contribution to the connective tissues of the periocular region and suggest that at their origin in the base of the skull, the EOMs and their associated connective tissues might have evolved neomorphically in situ from the same mesodermal source.

## RA responsiveness on the POM

Previous studies showed that NCCs have a critical role in the acquisition of cranial muscle morphology [9,13–16]. Our data using the *Tg:RARE-CreERT2* reporter and following invalidation of *Rarb/Rarg* in NCC derivatives suggest that the action of retinoic acid signaling on EOM patterning is also indirect, through its action on periocular connective tissues.

The observation that BMS493 treatments to impair ATRA signaling results in a more severe phenotype than that observed in *Aldh1a3* mutants could have several explanations. First, BMS493 is not a RAR antagonist but pan-RAR inverse agonist, which is capable of repressing RAR basal activity by favoring the recruitment of corepressors and promoting chromatin compaction [87]. Second, additional sources of ATRA could signal to the POM and be inhibited by the BMS493 treatment. Candidates include *Aldh1a1* (retina), *Aldh1a2* (temporal mesenchyme), and *Cyp1B1*, a member of the cytochrome p450 family of monooxygenases that can generate ATRA in the retina in a retinaldehyde dehydrogenease-independent manner [88]. Noneless, the severe patterning phenotypes observed in the *Aldh1a3* single mutant suggest that the local synthesis of ATRA in the RPE, optic stalk, and ventral retina at early stages (around E10.5) cannot be fully compensated by the action of other retinaldehyde dehydrogeneases in adjacent tissues (this study, [31,64]).

Interestingly, although the constitutive *RARE-hsp68-lacZ* transgene displays strong reporter activity in the retina and RPE [34,89], we observed a strong ATRA response in the choroid and in POM connective tissues with the *Tg:RARE-CreERT2;R26^{Tom}* reporter. However, one needs to consider that this type of reporter provides an off/on readout whose threshold level of response might depend on Cre levels, accesibility of the locus to recombination, and stochastic epigenetic mechanisms [90–93]. Given the difficulties in invalidating retinoic acid signaling in a cell-type-specific fashion and on cells perceiving a certain level of ATRA activity, it remains unclear how a gradient-like signaling system transitions to defined territories of transcription factor activity that will ultimately govern morphogenesis at a local level. We foresee that advances in single-cell transcriptomic approaches will help resolve the consecutive steps of musculoskeletal patterning in this and other anatomical locations.

## EOM and tendon patterning are concerted processes under RA control

We identified a short temporal window (E10.5–E11.75) in which ATRA is essential for the precise morphogenesis of several elements of the POM, including the formation of tendon

condensations and insertion sites in the POM-ring. These phenotypes are already evident from E11.75, a stage in which the EOM still figures as a muscle anlage. Interestingly, analysis of chick and mouse limb development also identified a short time window in which limb mesenchyme perturbations result later in muscle and tendon patterning defects [71,94]. Moreover, concomitant muscle and tendon defects have been observed in the chick limb and zebrafish jaw upon retinoic acid signaling misregulation [15,95]. Altogether, these results suggest that at certain anatomical locations, including the EOMs, endogenous local variations in the concentration of retinoids contribute to the establishment of *Scx-GFP+* condensations, a critical step in muscle functional unit assembly.

## Role of RA in the integration of EOM patterning and insertion site formation

The EOMs insert into the sclera, a non-bone structure in mammals. The generation of 4 attachment points that precisely mirror the position of the 4 recti muscles is a morphogenesis conundrum, as details on their specification are scarce. Four mesenchymal condensations have been described at the periphery of the developing optic cup in cat and human embryos, in positions facing the locations of the recti muscles [96,97]. In the mouse, at equivalent developmental timepoints (E11.5-E12.5) and homologous positions, apoptotic foci were observed and suggested to provide attachment for the recti muscles [51]. Using 3D imaging, we showed that at E11.75, the apoptotic foci are aligned with *Scx-GFP+* condensations that project medially, before any sign of splitting of the EOM anlage. As the apoptotic foci decrease in size from E12.5 onwards concomitant with ongoing muscle splitting and tendon formation, it is possible that they initially presage the POM for future tendon insertions but are not required for attachment per se. These foci appear in condensed SOX9+ PITX2+ cell domains of the future sclera; however, to our knowledge, their presence has not been reported in superstructures at other anatomical locations.

In agreement with the role of RA in the induction of cell death in other locations in the embryo [64], our study shows that upon perturbation of RA signaling, the apoptotic foci in the POM are absent or severely reduced. However, it is also possible that the loss of apoptotic foci upon RA inhibition is secondary to loss or mispatterning of the SOX9+ PITX2+ POM-ring. Thus, it will be of interest to determine whether null or NCC-specific mutations in PITX2 affect the patterns or amount of naturally occurring cell death in the periocular condensations. Although our work sheds light on the genetic mechanisms that regulate EOM insertion, the cellular mechanisms underlying this process remain undefined. It is tempting to speculate that the apoptotic foci are related to the disappearance of signaling centers [98,99] or compaction-mediated cell death and extrusion [100]. Future experiments directed towards modifying the amount or timing of cell death will be informative.

Few reports have demonstrated that mispatterning of specific muscles is coupled with aberrant superstructures [13,70,72] or removal of the prospective tendon attachment sites [95]. Our results are in agreement with that model, given that EOM mispatterning is concomitant with aberrant EOM insertions. As mammals do not develop a cartilage layer within the sclera [57,101], transient expression of SOX9 in the POM-ring of mouse embryos at the time of EOM patterning is intriguing. In the developing limb and jaw, tendon and bone are attached by a transitional connective tissue that develops from bipotent progenitors that co-express Scx and SOX9 [53–56], before progenitors are allocated to either cartilage or tendon lineages. Similarly, we observed a transient population of *Scx-GFP*/SOX9 double-positive cells at the insertion sites in the POM. In this context, our data suggest that SOX9 expression may represent a redeployment of the developmental module for tendon attachment, despite the fact that there

is no definitive cartilage in the mammalian sclera. Genetic studies will be required to ultimately assess the functional relevance of this population in EOM attachment.

Finally, transcriptome analysis revealed the existence of global and regional regulatory modules for superstructure patterning in the limb bones, offering a mechanism to induce variations in attachment sites without having to rewrite the entire skeletogenic program [102]. As several markers of the POM (*Pitx2*, *Foxc1/2*) and retinoic acid signaling modulators (*Cyp26a1/b1*) were identified as part of specific limb superstructure signatures [102], it is tempting to speculate that those genes also play a conserved role in the generation of the attachment module of the EOMs.

## Conclusion

The developing eye has been proposed to be key organizer of craniofacial development, independent of its role in vision [103]. This notion is based on the role of the eye for proper NCC migration to the periocular region [104], the common association of ocular and craniofacial developmental abnormalities [105] and the initial development of an eye in blind vertebrate species [103,106,107]. Moreover, as the appearence of a "camera-type" eye is a vertebrate innovation [108], and the EOM are already present in lamprey [4], it is possible that muscles and their target tissue might have co-evolved. Here, by characterizing coordinated patterning of the EOMs, their respective tendons and insertions, our findings illustrate further the role of the developing eye as a signaling center allowing integration of the EOM functional unit in the POM. Our results show that the tissue interactions during the development of this craniofacial muscle unit share features with those described in the limb but with additional regional properties (e.g., cell death, specification of attachments in the sclera) that seem to have been specifically incorporated into this group. Moreover, the capacity to instruct muscle patterning through variations in connective tissue derivatives provides a mechanism to explain the plasticity of the musculoskeletal system, at the anatomical and interspecies levels, while ensuring functional integration during evolution. These findings also imply that the generation of musculoskeletal units do not require major restructuring of the developmental programs of all the tissues implicated. Instead, co-option of a general program and simultaneous addition of local features appear to elaborate musculoskeletal diversity.

## Materials and methods

### Mouse strains and animal information

Animals were handled as per European Community guidelines, and the ethics committee of the Institut Pasteur (CETEA) approved protocols (Licence 2015–0008). The following strains were previously described: $Aldh1a3^{KO}$ [64], $Mesp1^{Cre}$ [46], $Tg(RARE-Hspa1b-cre/ERT2)$, designated here as $Tg:RARE-CreERT2$ [73], $R26^{Tom}$ (Ai9; [74]), $R26^{mTmG}$ [75], $Tg:Scx-GFP$ [48], $Myf5^{nlacZ}$ [109], $Myod^{iCre}$ ([79,110]), $Rarb^{flox}$ [111], $Rarg^{flox}$, [112], $Rdh10^{KO}$ [65], $Pax6^{Sey/Sey}$ [59], and $Tg:Wnt1^{Cre}$ [45]. $R26^{RAR403}$ ([78]) mice contain a loxP-flanked STOP sequence upstream of a mutated human RAR alpha gene, which behaves as a dominant negative receptor for all nuclear receptors upon Cre-mediated recombination. $Rdh10^{KO}$ embryos were received from the laboratory of Pascal Dollé. $Tg:Wnt1Cre;Rarb^{fl/fl};Rarg^{fl/fl}$ embryos were received from the laboratory of Valérie Dupé. $Pax6^{Sey/Sey}$ and control embryos were received from the laboratory of James Briscoe. $Tg:Lhx2^{Cre};Lhx2^{fl/fl}$ [61] were received from the laboratory of Leif Carlsson. $Shh(invC-6)2$ embryos were received from the laboratory of François Spitz.

To generate experimental embryos for $Mesp1^{Cre}$ or $Tg:Wnt1^{Cre}$ together with $Tg:Scx-GFP$ and $R26^{Tom}$ lineage tracings, Cre/+ males were crossed with $Tg:Scx-GFP;R26^{Tom/Tom}$ females. Mice were kept on a mixed genetic background C57BL/6JRj and DBA/2JRj (B6D2F1, Janvier

Labs). Mouse embryos and fetuses were collected between E10 and E18.5, with noon on the day of the vaginal plug considered as E0.5. Pregnant females were euthanized by cervical dislocation.

To induce recombination with the *Tg:RARE-CreERT2;R26*$^{mTmG}$ line, 5 mg of tamoxifen (Sigma #T5648) were administered by gavage to pregnant females. A 25 mg/ml stock solution in 5% ethanol and 95% sunflower seed oil was prepared by thorough resuspention with rocking at 4˚C.

To inhibit retinoic acid signaling, pregnant females of relevant genotypes were injected intraperitoneally with 10 mg/kg of BMS493 (Tocris, 3509), a pan-RAR inverse agonist. A 5-mg/ml BMS493 stock solution in DMSO (SIGMA, D2650) was prepared and stored in single use aliquots at −20˚C in tight cap tubes. At the time of injection, the aliquot was thawed, 200 μl of sterile PBS was added per 50μl aliquot and injected immediately.

## Immunofluorescence, detection of cell death, and in situ hybridization

Embryos were fixed for 2.5 hours in 4% paraformaldhehyde (PFA; 15710, Electron Microscopy Sciences) in PBS with 0,2–0,5% Triton X-100 (according to the embryonic stage) at 4˚C and washed overnight at 4˚C in PBS. For cryosectioning, embryos were equilibrated in 30% sucrose in PBS overnight at 4˚C and embedded in OCT. Cryosections (16–18 μm) were allowed to dry at RT for 30 minutes and washed in PBS. Immunostaining was performed as described in [113]. An anti-DsRed antibody (rabbit) was used to enhance the *R26*$^{Tom}$ signal except when co-staining was performed with antibodies raised in rabbit. In this case, the endogenous reporter signal was used. An anti-GFP antibody (chicken) was used to enhance the *R26*$^{mTmG}$ and *Tg:Scx-GFP* signal in whole-mount and section immunostainings.

*Scx* in situ hybridization was performed as per manufacturer instructions using the RNAscope Multiplex Fluorescent V2 Assay [113] and RNAscope Probe-Mm-Scx probe (Cat No. 439981). Sample pre-treatments were performed as described in [113]. Signal development was carried out using Opal 570 Reagent Pack (FP1488001KT, Perkin Elmer) diluted 1:1,500 in the ACD-provided TSA buffer and followed up by immunostaining.

TUNEL staining, which marks double-strand breaks, was performed with the In Situ Cell Death Detection Kit/Fluorescein (Roche, 11 684 795 910). Slides were first pretreated with a 2:1 mix of Ethanol:Acetic Acid for 5 minutes at −20˚C, washed twice for 20 minutes with PBS at RT, and processed for TUNEL staining as described by the manufacturer.

For whole-mount immunostaining, embryos were fixed and washed as described here and dehydrated in 50% Methanol in PBS and twice in 100% Methanol, 30 minutes each at RT and kept at −20˚C till needed. Heads were rehydrated, the periocular region was microdissected in PBS, and immunostaining performed as described in [113]. For embryos older than E13.5, the alternative pretreatment (containing 0.1% Tween-20, 0.1% TritonX100, 0.1% Deoxycholate, 0.1% NP40, 20% DMSO in PBS) and primary antibody immunolabeling steps of the idisco protocol (https://idisco.info/idisco-protocol/) were generally used. In all cases, secondary antibodies were applied in blocking buffer as described in [113] for >4 days at 4˚C with rocking. After immunolabelling, samples washed in 0.1% Tween/PBS, dehydrated in 50% Methanol in PBS and 100% Methanol 10 minutes each at RT, cleared with a mix benzyl alcohol and benzyl benzoate (BABB), and mounted for imaging as described in [114].

LysoTracker Red staining was used for detection of cell death in whole-mount live tissues as it reveals lysosomal activity correlated with increased cell death [52,115]. Embryos were quickly dissected in HBSS (Invitrogen, 14025–092), incubated in 2-ml tubes containing 5 μM of LysoTracker Red DND-99 (Molecular Probes, L7528) 45 minutes at 37˚C with rocking in

the dark, washed twice in PBS, fixed and processed for cryosections or whole-mount immunostaining as described previously.

## Antibodies

Primary and secondary antibodies used in this study are listed in S2 Table. To detect differentiating EOMs, we used a-smooth muscle actin (SMA), which is transiently expressed in differentiating myoblasts and myotubes [116–118]; Desmin, an early cytoskeletal muscle protein expressed in myoblasts, myotubes, and myofibers [116,119]; and myosin heavy chain (MyHC) to label sarcomeric myosin [120].

## Static imaging

A Zeiss SteREO Discovery V20 macroscope was used for imaging the endogenous fluorescence of whole embryos at the time of dissection. For tissue sections and whole-mount immunostaining of cleared embryos, a LSM700 and a LSM800 laser-scanning confocal microscope with ZEN software (Carl Zeiss, www.zeiss.com) were used.

All images were assembled in Adobe Photoshop and InDesign (Adobe Systems). Volume-3D rendering of the Z-stack series was performed in Imaris (version 7.2.1) software (Bitplane). For ease of EOM visualization, the Z-stack volumes were first manually segmented to define the EOM or the whole POM area using the Isosurface Imaris function. The signal outside the isosurface was set to zero, the corresponding channel duplicated, and subsequently, a new isosurface was created using automatic thresholding on the new channel. This new isosurface was used to calculate the corresponding EOM volumes.

## In situ hybridization

Whole-mount in situ hybridization with digoxigenin-labeled antisense mRNA probes was performed as described previously in [90]. The Aldh1a1, Aldh1a2, and Aldh1a3 probes were previously described in [34,36].

## micro-CT analysis

The tissue contrasting protocol has been adapted from the original protocol developed by [121] and applied to mouse embryos as described in [122] and [86]. For tissue contrasting, E13.5 embryos were stained in 0.5% phospho-tungstic acid (PTA) in 90% methanol for 4 days, E15.5 embryos were stained in 0.7% PTA in 90% methanol for 1 week.

The micro-CT analysis of the embryos was conducted using the GE phoenix v|tome|x L 240 (GE Sensing and Inspection Technologies GmbH, Germany), equipped with a 180 kV/15W maximum power nanofocus X-ray tube and flat panel detector DXR250 with 2048 × 2048 pixel, 200 × 200 μm pixel size. The exposure time of the detector was 900 milliseconds in every position over 360˚. Three projections were acquired and averaged in every position for reduction of the noise in micro-CT data. The acceleration voltage was set to 60 kV and tube current to 200 μA. The radiation was filtered by 0.2 mm of aluminium plate. The voxel size of obtained volumes (depending on a size of an embryo) appeared in the range of 2–6 μm. The tomographic reconstructions were performed using GE phoenix datos|x 2.0 3D CT software (GE Sensing and Inspection Technologies GmbH, Germany). The EOM, eye, and cartilages in the embryo head were segmented by an operator with semiautomatic tools within Avizo - 3D image data processing software (FEI, USA). The 3D segmented region was transformed to a polygonal mesh as an STL file and imported to VG Studio MAX 2.2 software (Volume Graphics GmbH, Germany) for surface smoothing and 3D visualization.

## Cell isolation from the periocular region and bulk cell cultures

The periocular region of $Tg:RARE-CreERT2;R26^{mTmG}$ embryos (including the eye itself) was microdissected and minced with small scissors inside a 2-ml Epperndorf tube. Samples were incubated with 1 ml of TrypLE Express (Invitrogen, 12604013) for 15 minutes at 37°C with agitation. Samples were resuspended by gently pipetting up and down 10–15 times using a P1000 pipette. Upon addition of 1 ml of culture media containing (10 μg/ml) DNaseI (Roche, 11284932001), samples were spun 15 minutes at 500$g$ at RT, pellet resuspended in 400 μl of culture media containing of 20% fetal bovine serum (FBS, Gibco), 1% Penicillin-Streptomycin (15140, Gibco), and 2% Ultroser G (15950–017, Pall Biosepra) in 50:50 DMEM:F12 (31966 and 31765, Gibco) and plated on individual wells of 8-well glass-bottom dishes (Ibidi, 80826) coated with 1 mg/ml of Matrigel (354234, BD Biosciences). Cells were allowed to attach for 8 hours at 37°C 5% $CO_2$, washed with PBS, and fixed for 15 minutes at RT with 4% PFA in PBS. After fixation, cells were washed in PBS and permeabilized with 0.5% Triton X-100 in PBS for 5 minutes at RT. After 3 washes in PBS (5 minutes each), cells were blocked with 20% goat serum in PBS 1 hour at RT. Primary antibodies were added to cells in 2% goat serum in PBS for 2 h at RT or ON at 4°C. Cells were washed 3 times with PBS, incubated with secondary antibodies for 1 hour at RT, washed in PBS, and kept in PBS for imaging.

## Statistics

The number of embryos of each genotype used for analysis is indicated in the figure legends and S1 Table. The graphs were plotted, and statistical analyses were performed using Prism8 (GraphPad Software, Inc). All data points are presented as mean ± SEM (error bars). Individual values can be found in S1, S2 and S3 Data files. Statistical tests used for analysis are indicated on the respective figure legends. $p$-values less than 0.05 were considered significant ($^{*}p < 0.05$; $^{**}p < 0.01$; $^{***}p < 0.001$).

## Supporting information

**S1 Fig. Lineage contributions to the EOM functional unit. (A-D)** NCC ($Tg:Wnt1^{Cre};R26^{Tom}$; $Scx$-$GFP$) and mesoderm ($Mesp1^{Cre};R26^{Tom};Scx$-$GFP$) lineage contributions to the periocular region on coronal cryosections of E17.5 embryos, combined with immunostaining for tendon (GFP) and muscle (Tnnt3, Troponin T Type 3, differentiated muscle). Sections at ventral (**A, C**) and dorsal (**B,D**) levels. Note that tendon insertions at the level of the orbit are NCC-derived (**A**), whereas the tendon origin (**D**) is mesoderm-derived. Higher-magnification views at the level of the tendon insertion (**A1,C1**) and origin (**B1,D1**) are shown as insets. Asterisk in **D1** indicates $Scx$-$GFP$+ cells in the perichondrium of the hypochiasmatic cartilage. (**E-F'**) NCC ($Tg:Wnt1^{Cre};R26^{Tom}$) and mesoderm ($Mesp1^{Cre};R26^{Tom}$) lineage contributions to the periocular region on coronal cryosections of E13.5 embryos, combined with immunostaining for TCF4 and muscle (PAX7/MYOD/MYOG, myogenic markers). Note that TCF4 is expressed robustly in connective tissue fibroblasts and at lower levels in myogenic cells. White arrowheads in (**E',F'**) mark Tom+, TCF4+, myogenic marker-negative connective tissue cells in the NCC- and mesoderm-derived areas. Pink arrowheads in (**E',F'**) mark myogenic cells (TCF4$^{low}$). (**G-H'**) Coronal cryosections of E13.5 $Tg:Scx$-$GFP$ embryos, combined with immunostaining for TCF4 (connective tissue, myogenic progenitors) and muscle (PAX7/MYOD/MYOG). White arrowheads in (**G',H'**) mark Tom+, TCF4+, myogenic marker-negative connective tissue cells in the NCC and mesoderm-derived areas. Pink arrowheads in (**G',H'**) mark myogenic cells (TCF4$^{low}$). (**I,J**) In situ hybridization on E12.5 (**I**) and E13.5 (**J**) coronal cryosections for $Scx$ combined with immunofluorescence for muscle (PAX7/MYOD/MYOG). High levels of $Scx$ mRNA are seen at the tendon origin and insertion but also in the bulk of the

muscle masses (**I',J'**, insets). a, anterior; d, dorsal; l, lateral; m, medial; MCT, muscle connective tissue; E, embryonic day; eom, extraocular muscle; NCC, neural crest cell; p, posterior; ti, tendon insertion; to, tendon origin; v, ventral.
(TIF)

**S2 Fig. Expression of PITX2 in the POM. (A-C)** Immunostaining on E11.5 (A), E12.5 (B) and E13.5 (C) coronal sections of control embryos for PITX2 (EOM myogenic progenitors, POM) and SMA (differentiated muscle). Arrowheads mark the EOM masses and asteriks point to PITX2 expression in the medial POM. Note thinning of the lateral POM as development proceeds (brackets). (*n* = 3 per stage). E, embryonic day; EOM, extraocular muscle; POM, periocular mesenchyme.
(TIF)

**S3 Fig. Developmental timing of EOM functional unit components. (A-D')** Immunostaining for the indicated markers on coronal sections of E11.5 (**A**), E12.5 (**B, E, F**), E13.5 (**C**) and E14.5 (**D**) *Tg:Scx-GFP* embryos. (**A'-D"**) Higher-magnification views of the anterior tendon insertions (superior rectus) in the POM. White arrowheads mark SOX9+ *Scx-GFP*+ cells up to E13.5. Asterisk in **A'-C"** point to SOX9+ *Scx-GFP*-negative areas. Double asterisk in **D'** mark *Scx-GFP*+ SOX9-negative tendon tips at E14.5. SOX9 expression remains at the sclera. (**E-F'**) Immunostaining for the indicated markers on coronal sections of E11.75 and E13.5 *Tg:Scx-GFP* embryos preincubated with LysoR. Arrowheads in **E,E'** indicate SOX9+ LysoR+ cells in the E11.75 POM. (**G-H"**) Immunostaining for the indicated markers on tranversal sections of E12.5 *Tg:Scx-GFP* embryos. (**G'-H"**) Higher magnification views of the lateral tendon insertions (lateral rectus) showing SOX9+ or PITX2+ LysoR+ areas. MYOD/MYOG (myogenic markers) in **A-D** and MyHC (myofibers) in **G-H** were used to identify the EOMs. (*n* = 3 per stage). a, anterior; d, dorsal; E, embryonic day; EOM, extraocular muscle; l, lateral; LysoR, LysoTracker Red; m, medial; p, posterior; POM, periocular mesenchyme; s, sclera; v, ventral.
(TIF)

**S4 Fig. EOM morphogenesis in mutants with eye defects. (A-C)** Micro-CT-based 3D reconstruction of EOM, eyeball, optic nerve, and lens in E13.5 control (**A**), *Pax6^{Sey/+}* (**B**), and *Pax6-^{Sey/Sey}* embryos (**C**). Note that in heterozygote embryos, EOM patterning proceeds normally despite having a smaller retina and lens (*n* = 3). (**D-E**) Micro-CT-based 3D-reconstruction of EOM and eyeball in E18.5 control (**A**) and *Tg:Lhx2^{Cre};Lhx2^{fl/fl}* (**B**) embryos. Arrowheads highlight some extent of EOM segregation in the mutant. (**D'-E'**) Coronal sections of control (**D'**) and mutant (**E'**) embryos stained with MyHC (differentiated muscle). Arrowheads indicate individual EOM masses (*n* = 2). (**F-F"**) Analysis of EOM patterning in E15.5 embryos containing inversions of *Shh* genomic regulatory regions (Inv(6-C2)). (**F**) Skeletal preparation of mutant embryos displaying cyclopia. (**F'-F"**) Micro-CT-based 3D-reconstruction of EOM and eyeball of mutant embryos (*n* = 2). a, anterior; ctrl, control; d, dorsal; E, embryonic day; EOM, extraocular muscle; l, lateral; m, medial; micro-CT, micro-computed tomography; p, posterior; v, ventral.
(TIF)

**S5 Fig. EOM morphogenesis is dependent retinoic acid signaling. (A)** Whole-mount in situ hybridization for *Aldh1a1*, *Aldh1a2*, and *Aldh1a3* in E10.5, E11.5, and E12.5 wildtype embryos (*n* = 3). (**A'-A"**) In situ hybridization for *Aldh1a2* (**A'**) and immunostaining for ALDH1A2 and MyHC (differentiated muscle) (**A"**) on E12.5 coronal sections. ALDH1A2 is expressed in the temporal mesenchyme and adjacent connective tissues (*n* = 3). (**B**) Micro-CT-based 3D-reconstruction of eyeball, optic nerve and lens of E13.5 control, *Aldh1a3^{KO}* and BMS493-treated embryos (*n* = 2 each genotype). The lower row is a scheme of a sphere fitting the

eyeball and lens, and a cylinder for the optic nerve. Note ventralization of the eyeball in *Aldh1a3*[KO] and BMS493-treated embryos. **(C)** Micro-CT-based 3D-reconstruction of the mesenchymal condensations of the nasal capsule and trabecular cartilage of E13.5 control (white), *Aldh1a3*[KO] (yellow) and BMS493-treated (green) embryos. **(D-D')** MyHC WMIF of E13.5 BMS493-treated embryos as described in S1 Table. The most severe phenotype obtained in each condition is shown as an isosurface. Two different examples of the phenotype observed upon treatment VI (a,b) are shown (most and least severe). An *Aldh1a3*[KO] embryo is shown as comparison. Asterisk denotes ectopic duplicated SO muscle. **(D')** Higher-magnification views from **D** (dashed squares) with examples of adjacent, nonsplit muscle masses (1,2) depicted by differential fiber orientation. **(E-F)** MyHC WMIF of E13.5 control (**E**) and *Rdh10*[KO] (**F**) embryos. Note total absence of muscle splitting in mutant (*n* = 3 per condition). EOMs in **D-F** were segmented from adjacent head structures and 3D-reconstructed in Imaris (Bitplane). ctrl, control; dr, dorsal retina; 3D-rec, 3D-reconstruction; e, eyelid groove; E, embryonic day; EOM, extraocular muscle; io, inferior oblique; ir, inferior rectus; le, lens; lgb, lacrimal gland bud; lr, lateral rectus; micro-CT, micro-computed tomography; mr, medial rectus; nlg, nasolacrimal groove; np, nasal pit; os, optic stalk; rb, retractor bulbi; rpe, retinal pigmented epithelium; so, superior oblique; sr, superior rectus; tm, temporal mesenchyme; vr, ventral retina, WMIF, whole-mount immunofluorescence.
(TIF)

**S6 Fig. Retinoic acid signaling responsiveness in the periocular region. (A-C)** Macroscopic views of endogenous GFP fluorescence of *Tg:RARE-CreERT2;R*[26mTmG] embryos. Tam was injected into pregnant females and embryos analyzed at indicated time points. Arroheads indicate labeling in POM (*n* > 3 per condition). **(D)** Immunostaining on cells isolated from the periocular region of *Tg:RARE-CreERT2;R*[26mTmG] embryos for GFP and myogenic markers (MYOD/MYOG). **(E)** Strategy used to determine responsiveness of *Tg:RARE-CreERT2* reporter in presence of BMS493. BMS493 was injected to pregnant females every 10–12 hours between E10 and E11.75. Recombination was induced by tamoxifen at E10.5 (2 hours after the first BMS injection). **(F-I)** WMIF for MyHC (differentiated muscle) and GFP (ATRA-responsive cells) of control (**F,G**) and BMS493-treated embryos (**H,I**). BMS493 treatment before and after tamoxifen induction reveals a drastic decrease in GFP+ cells in the periocular region (**I**) compared with controls (**G**). Asterisks mark the location of the optic nerve (*n* = 3). **(J)** Coronal sections (dorsal, EOM origin) of E12.5 *Tg:RARE-CreERT2;R26*[Tom]*;Scx-GFP* embryos immunostained for GFP, Tom (ATRA-responsive cells) and MYOD/MYOG (muscle). Higher magnification views as insets. Arrowheads in J1 mark Tom-negative myogenic cells, and asterisks indicate sporadic labeling in myogenic cells. Arrowheads in J2 mark Tom+ *Scx-GFP+* cells (*n* = 3). **(K-L)** WMIF for SMA (differentiated muscle) of Myod[iCre] (control, **K, K'**) and *Myod*[iCre]*;R26*[RAR403] (mutant embryos, **L,L'**) (*n* = 3). EOMs in **F-I** and **K-L'** were segmented from adjacent head structures and 3D-reconstructed in Imaris (Bitplane). a, anterior; ATRA, all-trans retinoic acid; ctrl, control; d, dorsal; E, embryonic day; EOM, extraocular muscle; l, lateral; m, medial; p, posterior; POM, periocular mesenchyme; Tam, tamoxifen; v, ventral; WMIF, whole-mount immunofluorescence.
(TIF)

**S7 Fig. EOM insertions in the POM are altered upon ATRA deficiency. (A-D')** Immunostaining for the indicated markers on coronal E12 sections of control (**A,C,C'**) and BMS493-treated (**B,D,D'**) *Tg:Scx-GFP* embryos pre-incubated with LysoTracker Red (LysoR). **(C',D')** Higher-magnification views of the POM region. In BMS493-treated embryos, LysoR and SOX9 staining are absent in the POM (**D'**, asterisk) and SOX9 staining also missing in the RPE (**D'**, double asterisk). **(E-N)** WMIF for the indicated markers of E11.75 control (**E,F,K**),

*Aldh1a3$^{KO}$* (**G,H,M**) and BMS493-treated (**I,J,N**) embryos pre-incubated with LysoR (right eyes). In lateral views, neither *Aldh1a3$^{KO}$* or BMS493-treated embryos (**H,J**) show a full PITX2+ POM-ring as the controls (**F**). Arrowheads in (**G,H**) mark remaining expression of SOX9/PITX2 in the POM of mutant or inhibitor treated embryos. Asterisks in (**F,H,J**) mark apoptosis spots in the POM. (**K-N**) Segmented medial views of periocular region of control, *Aldh1a3$^{KO}$* or BMS493-treated embryos. Volumes were truncated in Z for clarity. Full view as schemes below. The PITX2+ POM-ring is continuous with the medial-POM in control embryos (**K**). In mutant and BMS493-treated embryos, residual PITX2 expression in POM is discontinuous with the medial-POM (blue arrowheads) (**M,N**). (**O-Q'**) WMIF for SMA (differentiated muscle), GFP (ATRA-responsive cells) and PITX2 (muscle progenitors, POM) of *Tg:RARE-CreERT2;R26$^{mTmG}$* control (**O,O'**), *Aldh1a3$^{KO}$* (**P,P'**) and BMS493-treated (**Q,Q'**) embryos. Blue arrowheads in **P** and **Q** show reduction or loss of ATRA-responsive cells in the periocular region. (**O'-Q'**) Single Z-planes of the segmented volume. White arrowhead in **O** shows correct PITX2+ connective tissue pre-pattern in the prospective muscle areas and arrow marks PITX2 expression along the muscle masses. Blue arrowheads in **P'** and **Q'** show reduction or loss of ATRA-responsive cells and PITX2 pre-pattern in the medial periocular mesenchyme. (**R-T**) Single cell plane of WMIF of E12 control (**R**), BMS493-treated (**S**) and *Myf5$^{nlacZ/nlacZ}$* embryos (**T**) (right eyes) for TCF4 (muscle connective tissue) and SMA (differentiated muscle). Asterisks mark position of the optic nerve. White arrowheads in **R,T** show correct TCF4+ connective tissue pre-pattern in the prospective muscle areas and white arrow marks TCF4 expression along the muscle masses (**R**). (**U-V**) WMIF for GFP and β-gal (myogenic progenitors) of E14.5 *Tg:Scx-GFP;Myf5$^{nlacZ/+}$* (**U**, control) and *Tg:Scx-GFP;My5$^{nlacZ/nlacZ}$* (**V**, mutant) embryos (left eyes). Asterisk indicates few remaining β-gal+ cells in mutant. (**U', V'**) split GFP channel. Arrowheads in (**V**) indicate the correct position of tendon condensations for the 4 recti muscles although these are absent in the mutant. Lower panel, higher-magnification views. a, anterior; ATRA, all-trans retinoic acid; anl, anlage; apop, apoptosis; ctrl, control; d, dorsal; E, embryonic day; eom, extraocular muscle; l, lateral; m, medial; m-pom, medial periocular mesenchyme; l-pom, lateral periocular mesenchyme; POM, periocular mesenchyme; pom-r, periocular mesenchyme ring; p, posterior; rpe, retinal pigmented epithelium; v, ventral; WMIF, whole-mount immunofluorescence.
(TIF)

**S8 Fig. Illustration depicting the timeline of EOM and POM development in relation to ATRA signaling.** Summary of the most relevant genetic perturbations and drug treatments of this study. ATRA, all-trans retinoic acid; EOM, extraocular muscle; POM, periocular mesenchyme.
(TIF)

**S1 Table. BMS493 injection regimes.** The same BMS493 concentration (Conc BMS) was used across injection regimes, from I to VI. Pink cases mark injection time points in the morning (mo), midday (mid) or evening (eve) between E10.5 and E12.5. The number of embryos with different muscle phenotype severity at E13.5 are shown: (**no**), no altered phenotype; mild (**+**), muscle mispatterning but overall organization retained; strong (**++**), strong mispatterning but with a minimum of 2 muscles split; severe (**+++**), no or almost absent splitting. n is the total number of embryos analyzed for a certain treatment type (*n* = 38 embryos analyzed in total); N exp, number of times the experiment was repeated. E, embryonic day; EOM, extraocular muscle.
(XLSX)

**S2 Table. Antibodies and resources used in this study.**
(XLSX)

**S1 Data. Excel table containing individual values to generate the histogram in Fig 3K.**
For each tissue section, a rectangular ROI (region of interest) was defined in Fiji (Image J) in
the LysoTracker Red-positive and adjacent LysoTracker Red-negative areas. The number of
SOX9+ cells was scored in each ROI. Individual values were represented by the ratio of
N˚cells/ROI area (μm2).
(XLSX)

**S2 Data. Excel table containing individual values to generate the histogram in Fig 4I.** For
each embryo, the EOM volume was calculated as described in the methods section. The nor-
malized EOM volume is the ratio of the individual volume value/average of the control values.
EOM, extraocular muscle.
(XLSX)

**S3 Data. Excel table containing individual values to generate the histogram in Fig 5C.** Each
data point represents the percentage of myogenic and non-myogenic GFP+ cells isolated by
bulk digestion of the periocular area of a single *Tg:RARE-CreERT2;R26^{mTmG}* reporter embryo.
(XLSX)

**S1 Video. Temporal sequence of EOM patterning.** Whole-mount immunostaining for
MYOD/MYOG/Desmin (E11.5 to E12.5) or MyHC (E13.5 to E14.5) on the developing EOM
of control embryos. EOMs were segmented from adjacent head structures and 3D-recon-
structed in Imaris (Bitplane). E, embryonic day; EOM, extraocular muscle.
(MOV)

**S2 Video. Organization of the EOM functional unit at E11.75.** Whole-mount immunostain-
ing for MyHC (myofibers, Red) and PITX2 (POM and myogenic progenitors, Cyan) on con-
trol embryo. EOMs were segmented from adjacent head structures and 3D-reconstructed in
Imaris (Bitplane). E, embryonic day; EOM, extraocular muscle; POM, periocular mesen-
chyme.
(MOV)

**S3 Video. Organization of the EOM functional unit at E11.75.** Whole-mount immunostain-
ing for MyHC (myofibers, Cyan) and GFP (tendon progenitors, Green) on *Tg:Scx-GFP*
embryo. EOMs were segmented from adjacent head structures and 3D-reconstructed in Imaris
(Bitplane). Apoptotic foci are visualized by LysoTracker Red staining. A clipping plane was
added for clarity of visualization. Note that tendon condensations start to organize radially
and towards the apoptotic foci in the POM. E, embryonic day; EOM, extraocular muscle;
POM, periocular mesenchyme.
(MOV)

**S4 Video. Organization of the EOM functional unit at E12.5.** Whole-mount immunostain-
ing for MyHC (myofibers, Cyan) and GFP (tendon progenitors, Green) on *Tg:Scx-GFP*
embryo. EOMs were segmented from adjacent head structures and 3D-reconstructed in Imaris
(Bitplane). Apoptotic foci are visualized by LysoTracker Red staining. A clipping plane was
added for clarity of visualization. Note how tendon condensations of the recti muscles are
more refined at this stage and project towards the apoptotic foci in the POM. E, embryonic
day; EOM, extraocular muscle; POM, periocular mesenchyme.
(MOV)

**S5 Video. Organization of the EOM functional unit of E13.5 BMS493-treated (E10.5 →
E11.75) *Tg:Scx-GFP* embryos.** Whole-mount immunostaining for MyHC (myofibers, Red)
and GFP (tendon progenitors, Green). EOMs were segmented from adjacent head structures

and 3D-reconstructed in Imaris (Bitplane). Note the presence of tendon condensations at the tips of mispatterned muscle masses. E, embryonic day; EOM, extraocular muscle.
(MOV)

**S6 Video. ATRA-responsiveness in the periocular region of E13.5 *Tg:RARE-CreERT2; R26^{mTmG}* control embryos (Tamoxifen induction at E10.5).** Whole-mount immunostaining for MyHC (myofibers, Red), and GFP (ATRA-responsive cells, Green). EOMs were segmented from adjacent head structures and 3D-reconstructed in Imaris (Bitplane). EOMs are shown as isosurfaces for clarity of visualization. ATRA, all-trans retinoic acid; E, embryonic day; EOM, extraocular muscle.
(MOV)

**S7 Video. ATRA-responsiveness in the periocular region of E13.5 BMS493-treated *Tg: RARE-CreERT2;R26^{mTmG}* embryos (BMS493 treatment: E10 → E11.75; Tamoxifen induction at E10.5).** Whole-mount immunostaining for MyHC (myofibers, Red) and GFP (ATRA-responsive cells, Green). EOMs were segmented from adjacent head structures and 3D-reconstructed in Imaris (Bitplane). EOMs are shown as isosurfaces for clarity of visualization. ATRA, all-trans retinoic acid; E, embryonic day; EOM, extraocular muscle.
(MOV)

**S8 Video. ATRA-responsiveness in the periocular region of E12.5 *Tg:RARE-CreERT2; R26^{mTmG}* control embryos (Tamoxifen induction at E10.5).** Whole-mount immunostaining for GFP (ATRA-responsive cells, Green), PITX2 (POM and myogenic progenitors, Cyan) and SMA (differentiated muscle, Red). Z-planes from medial to lateral levels. ATRA, all-trans retinoic acid; E, embryonic day; POM, periocular mesenchyme.
(MOV)

**S9 Video. ATRA-responsiveness in the periocular region of E12.5 *Tg:RARE-CreERT2; R26^{mTmG};Aldh1a3^{KO}* embryos (Tamoxifen induction at E10.5).** Whole-mount immunostaining for GFP (ATRA-responsive cells, Green), PITX2 (POM and myogenic progenitors, Cyan), and SMA (differentiated muscle, Red). Z-planes from medial to lateral levels. ATRA, all-trans retinoic acid; E, embryonic day; POM, periocular mesenchyme.
(MOV)

**S10 Video. ATRA-responsiveness in the periocular region of E12.5 *Tg:RARE-CreERT2; R26^{mTmG}* BMS493-treated embryos (BMS493 treatment: E10 → E11.75; Tamoxifen induction at E10.5).** Whole-mount immunostaining for GFP (ATRA-responsive cells, Green), PITX2 (POM and myogenic progenitors, Cyan) and SMA (differentiated muscle, Red). Z-planes from medial to lateral levels. ATRA, all-trans retinoic acid; E, embryonic day; POM, periocular mesenchyme.
(MOV)

**S11 Video. Organization of the EOM insertions of E12.5 *Tg:Scx-GFP* control embryos.** Whole-mount immunostaining for MyHC (myofibers, Red) and GFP (tendon progenitors, Green). EOMs were segmented from adjacent head structures and 3D-reconstructed in Imaris (Bitplane). A clipping plane has been added for clarity of visualization. E, embryonic day; EOM, extraocular muscle.
(MOV)

**S12 Video. Organization of the EOM insertions of E12.5 BMS493-treated (E10.5 → E11.75) *Tg:Scx-GFP* embryos.** Whole-mount immunostaining for MyHC (myofibers, Red) and GFP (tendon progenitors, Green). EOMs were segmented from adjacent head structures

and 3D-reconstructed in Imaris (Bitplane). A clipping plane has been added for clarity of visualization. Note that distribution of *Scx-GFP+* cells is more difuse than for controls (Video 11), but an *Scx-GFP+* condensation is present at the tip of the inferior muscle mass. E, embryonic day; EOM, extraocular muscle.
(MOV)

**S13 Video. Organization of the EOM insertions of E14.5 *Tg:Scx-GFP;Myf5$^{nlacZ/+}$* embryos.** Whole-mount immunostaining for β-gal (myogenic progenitors, Red) and GFP (tendon, Green). EOMs were segmented from adjacent head structures and 3D-reconstructed in Imaris (Bitplane). E, embryonic day; EOM, extraocular muscle.
(MOV)

**S14 Video. Organization of the EOM insertions of E14.5 *Tg:Scx-GFP;Myf5$^{nlacZ/nlacZ}$* embryos.** Whole-mount immunostaining for β-gal (myogenic progenitors, Red) and GFP (tendon, Green). E, embryonic day; EOM, extraocular muscle.
(MOV)

## Acknowledgments

We gratefully acknowledge the UtechS Photonic BioImaging (Imagopole), C2RT, Institut Pasteur. We thank Robert Kelly (IBDM, Marseille, France) for kindly providing *R26$^{RAR403}$* mice.

## Author Contributions

**Conceptualization:** Glenda Evangelina Comai, Shahragim Tajbakhsh.

**Data curation:** Glenda Evangelina Comai.

**Formal analysis:** Glenda Evangelina Comai, Markéta Tesařová.

**Funding acquisition:** Shahragim Tajbakhsh.

**Investigation:** Glenda Evangelina Comai.

**Methodology:** Glenda Evangelina Comai, Markéta Tesařová, Tomáš Zikmund, Jozef Kaiser.

**Project administration:** Glenda Evangelina Comai, Shahragim Tajbakhsh.

**Resources:** Glenda Evangelina Comai, Valérie Dupé, Muriel Rhinn, Pedro Vallecillo-García, Fabio da Silva, Betty Feret, Katherine Exelby, Pascal Dollé, Leif Carlsson, Brian Pryce, François Spitz, Sigmar Stricker, James Briscoe, Andreas Schedl, Norbert B. Ghyselinck, Ronen Schweitzer, Shahragim Tajbakhsh.

**Supervision:** Jozef Kaiser, Ronen Schweitzer, Shahragim Tajbakhsh.

**Validation:** Glenda Evangelina Comai.

**Visualization:** Glenda Evangelina Comai, Markéta Tesařová.

**Writing – original draft:** Glenda Evangelina Comai.

**Writing – review & editing:** Glenda Evangelina Comai, Markéta Tesařová, Valérie Dupé, Muriel Rhinn, Pedro Vallecillo-García, Fabio da Silva, Leif Carlsson, Sigmar Stricker, Tomáš Zikmund, Jozef Kaiser, James Briscoe, Andreas Schedl, Norbert B. Ghyselinck, Ronen Schweitzer, Shahragim Tajbakhsh.

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
