## [Editor Report · Decision Letter 0]

14 Jan 2020

Dear Dr Comai, 

Thank you for submitting your manuscript entitled "Local retinoic acid directs emergence of the extraocular muscle functional unit" for consideration as a Research Article by PLOS Biology.

Your manuscript has now been evaluated by the PLOS Biology editorial staff as well as by an academic editor with relevant expertise and I am writing to let you know that we would like to send your submission out for external peer review.

Please re-submit your manuscript within two working days, i.e. by Jan 16 2020 11:59PM.

Kind regards,

Ines

--

Ines Alvarez-Garcia, PhD

Senior Editor

PLOS Biology

Carlyle House, Carlyle Road

Cambridge, CB4 3DN

+44 1223–442810

---

## [Decision Letter · Decision Letter 1]

3 Mar 2020

Dear Dr Comai,

Thank you very much for submitting your manuscript "Local retinoic acid directs emergence of the extraocular muscle functional unit" for consideration as a Research Article at PLOS Biology. Thank you also for your patience as we completed our editorial process, and please accept my apologies for the delay in providing you with our decision. Your manuscript has been evaluated by the PLOS Biology editors, an Academic Editor with relevant expertise, and by four independent reviewers.

As you will see, all four reviewers are generally positive and find the conclusions of the manuscript interesting, however they also raise points about the choice of markers in some of the experiments, which we are concerned might, potentially, affect the interpretation of the results. They indicate that the manuscript would benefit from improvements to the figures to clarify the data and also to the text to make it more accessible to a wide audience.

In light of the reviews (attached below), we will not be able to accept the current version of the manuscript, but we would welcome re-submission of a much-revised version that takes into account the reviewers' comments. We cannot make any decision about publication until we have seen the revised manuscript and your response to the reviewers' comments. Your revised manuscript is also likely to be sent for further evaluation by the reviewers.

We expect to receive your revised manuscript within 2 months. 

**IMPORTANT - SUBMITTING YOUR REVISION**

*Re-submission Checklist*

*Published Peer Review*

*PLOS Data Policy*

*Blot and Gel Data Policy*

Sincerely,

Ines

--

Ines Alvarez-Garcia, PhD

Senior Editor

PLOS Biology

Carlyle House, Carlyle Road

Cambridge, CB4 3DN

+44 1223–446970

Reviewers’ comments

Rev. 1:

This is an extremely detailed analysis of the development of the extraocular eye muscles and the role of retinoic acid in their formation analysed by superb 3-D reconstructions, genetic lineage tracing, in situ hybridization and genetic knockouts. They show that the periocular mesenchyme is not exclusively the domain of the neural crest as previously though, but the mesenchyme itself plays a role. The role of retinoic acid is to act on the neural crest and its source is the eye itself. This is a fascinating piece of work relevant not only to retinoic ccid signaling but to embryology as a whole and to evolution. I have no comments of suggestions for improvement and recommend its acceptance.

Rev. 2:

The manuscript describes well-constructed and executed series of experiments that address an important and fundamental question of how individual muscle units are constructed. Specifically, they address the formation of the extraocular muscles.

A key message from Figure 1 is that EOMs have connective tissue contributions from two distinct embryonic origins-neural crest laterally and cranial mesoderm medially. I found the presentation and description of the results quite hard to follow, to the point that the central message was lost/ less convincing to some extent. Specifically, L166 expression of markers is said to delimit a triangular area-I don't see this (I do see a clearer triangular area in panels E-H but this is not commented on).

The Wnt1Cre lineage marker seems to label almost all cells in the sections shown. The region of muscle cells (labelled with Pax7/MyoD/Myogenin-shown in blue) appears negative -consistent with the muscle lineage being derived from cranial mesoderm but it is within the region that I assume we are supposed to also see mct (labelled as such in lower panel A) but this is not clear in this panel-a higher mag view could help.

The Scx-GFP that can be seen in this region in panel1B is not ideal as Scx is well established tendon marker and is expressed broadly in other regions of the section. It is problematic to use as an MCT marker in this context as the cells adjacent to the Muscle marker positive cells shown in panel A could be tendon cells rather than MCT. What could help here is some clear positive data showing co-labelling of Wnt1Cre R26Tom with an established mct marker. On a more minor point, I think it would be helpful to label lower panels A' etc to help the reader navigate the figure.

I think it would also help if for each data set using the NCC marker (Wnt1Cre) or mesoderm marker (Mesp1Cre) that comparable sections were shown with the reciprocal lineage marker ie. For panels A-B (ventral sections) that the staining obtained with Msp1Cre is shown and vice versa with panels E-H that the Wnt1Cre lineage marker is shown in the dorsal sections. This could help to clearly visualize the exclusion of cells of distinct embryonic origin to muscle origins and insertions. I found the schematic confusing/misleading as it shows two muscle domains when only one domain (posterior?) is visible in the IHC sections (Fig1A-D). I also found the key to the schematic hard to follow as the neural crest derived MCT is actually shown superimposed on the muscle and so appears as red dots rather than pink as shown in key

Another very minor point, dotted lines (167) are described as dashed (573) in figure legend. It is clearer for the reader if terminology is consistent.

Fig2 and 3 provide detailed description of the formation of the EOM and how this can be disrupted in mouse mutants where the eye is abnormally formed. This study describes the normal formation and abnormal formation of the EOMs at greater resolution than has been described previously.

The authors further use a novel RARE reporter combined with R26mTmG to track the lineage of RA-responsive cells. L329 onwards describe quantification of labelling and that minimal (co)labelling was detected in myogenic cells. It might be expected that the responsive cells are mct but this is not directly addressed.

This is analyzed further using Scx-GFP as a marker of both tendon and connective tissue progenitors (Fig6). This is again problematic/a little unsatisfying as it is hard to be sure if the GFP cells being studied are tendon precursors or truly connective tissue progenitors. The text is careful to not over-state the results but would it be possible to use markers to distinguish mct in these regions more directly

Further comments:

Figure 1 could include DAPI staining to distinguish non-neural crest and non-mesoderm derived cells.

Figure 1A bottom panel could show muscle staining to be consistent with the other bottom panels.

As stated above, the mct labelled in this panel is not obvious.

Figure 1B. corneal epithelium is double labelled?

There is no structure called "corneal ectoderm". There is a surface ectoderm. However, by E13.5 the surface ectoderm does not exist. Prior to this stage, the surface ectoderm has detached from the lens and becomes the corneal epithelium. You can see the "corneal epithelium" faintly as the most anterior layer in the images shown. The area referred to as the "corneal ectoderm" are the neural crest cells, which differentiate into the keratocytes to lay down the corneal stroma.

Labels are missing from some of the figures. E.g. Fig S4 SE? I also don't understand the CE label in this image?

Why is ALDH1A3 staining visible in the corneal epithelium at E11.5 but not at E12.5 in the controls?

SE in fig 4 should have an arrow to the outermost layer of cells. The SE is stained on the nasal side but not the temporal, has a layer of cells been lost?

Fig. 6 shows type XII collagen staining, interestingly there is also a loss of expression in the corneal epithelium?

MINOR TYPO

Line 80: space needed between NCC and independent

Line 85: "NCC cells" should be NCC

Line 244 "leds" should be leads?

Line 428 spelling "evolutionary"

L332 connective typo

Rev. 3:

This is a very interesting study that investigates developmental coordination of extraocular muscles (EOM), tendons, and their attachment sites in the sclera of the eye. Using genetic lineage analysis, the authors show that 1) the periocular connective tissue (muscle connective tissue and tendons) is of dual mesoderm/neural crest cell (NCC) origin and 2) the embryonic origin of the tendon insertion matches that of the muscle connective tissue. Using mouse genetics and inhibitor treatments, the authors go on to show that neural derivatives of the eye regulate morphogenesis of the EOM anlage and its scleral attachments by activating retinoic acid signaling in the NCC-derived periocular connective tissue. This study is exciting because it fills a gap in knowledge about the developmental mechanisms regulating tendon insertion into non-bone tissues. While I am enthusiastic about this work, I believe the study would benefit from some reworking (and possibly reorganization) of the results section to make it more accessible to the wide readership at PLOS Biology. In general, it was very hard to appreciate the significance of the results while keeping track of the many mouse mutants introduced.

Here are specific comments related to the results:

1. Since the embryonic lineage tracing data of the periocular connective tissue differs from what was previously published, it is important to show dorsal section of Wnt1-Cre and ventral sections of Mesp1-Cre at E13.5 in Figure 1 (the reciprocal sections in Supplemental Figure 1 are at E17.5).

2. It is not made clear in the main text why chondrogenic factor Sox9 is being used. Based on the figure legend, it seems as though it is being used as a marker for the dense fibrous connective tissue of the sclera. However, it could also be used to identify the tendon attachment sites.

3. The sclera is not labelled in the figures (1, 2 , etc.) despite being listed in the legend.

4. Line 179: Either show the E11.5 data or delete statement about data not shown.

5. Line 205: It is intriguing that apoptotic foci mark the future tendon attachment sites. What cells are undergoing apoptosis? It is difficult to see if these are Sox9/Pitx2 cells or another subpopulation of POM cells given the image magnification.

6. Line 230: It is concluded that Sox9 and Scx have complementary patterns at the EOM insertion sites. It is important to know if these regions of overlap are composed of Sox9/Scx double positive cells that form the tendon-bone attachments in the mouse limb and jaw because the presence of Sox9/Scx double positive cells will suggest conservation of tendon attachment progenitor cells and strengthen the argument that some aspects of tendon-bone attachment are conserved during tendon-sclera attachment.

7. Line 335: It is not made clear how the identity of the reporter-positive cells was determined to be connective tissue. Nor is it clear what is meant by "bulk cell preparations" in the main text or figure ledged. Immunofluorescence followed by cell sorting? What marker was used to identify connective tissue cells?

8. In Figure 5, it is shown that NCC-specific loss of Rarb/Rarg leads to aberrant muscle morphogenesis (failure of splitting). Since the muscle connective tissue and tendons around the eye are of dual NCC-mesoderm origin, is it believed that RA signaling is specific to the NCC-derived connective tissue? Is the phenotype only restricted to regions in which the periocular connective tissue is NCC-derived? Is RA signaling important in the mesoderm-derived periocular connective tissue?

9. Line 406: J' should be G'

10. A summary figure at the end would be very helpful for the reader.

Rev. 4:

In their manuscript, Comai and colleagues investigate the involvement of RA signaling in the genesis of extraocular muscle connections. They use mouse genetics and imaging to assess the developmental origins of the EOMs and their connective tissue. In contrast to other craniofacial muscle attachments, they find a dual neural crest cell and mesoderm origin to the connective tissue. They demonstrate that the EOMs form from a single anlage that subsequently splits. They show that attenuation of RA signaling either genetically or pharmacologically causes defects in the formation and insertion of the EOMs. They demonstrate that a short window of signaling (E10.75) is critical for the orientation of these muscle fibers and use a genetic reporter to characterize those cells that are RA-responsive. The schematics and supplemental movies that the authors use greatly assist in conveying complex anatomical details. The manuscript is well written and provides important new insights into an understudied aspect of biology. I have only a few concerns:

1) The authors state that Sox9 and Scx expression patterns were segregated at E13.5 (lines 220-221). This is very difficult to appreciate from the images where it looks as if there is some co-expression. Given that Sox9;Scx co-expressing cells have been described in other systems, it would be worthwhile for the authors to determine if there is co-expression in the EOM connective tissue.

2). For Figure 5, it would be more informative to show data from induction at E10.5 (rather than E 9.5), given that the authors show that E10.75 is a critical time window for RA signaling for EOM attachment. From Suppl Fig. 5, it seems that the authors must have these data. Adding some sections from this time point could be informative as to the subpopulation of RA-responsive cells most important in patterning the EOM.

3) Intro, line 86. McGurk et al. examined zebrafish not mice (the species is correctly stated in the discussion).

---

## [Decision Letter · Decision Letter 2]

19 Aug 2020

Dear Dr Comai,

Thank you for submitting your revised Research Article entitled "Local retinoic acid directs emergence of the extraocular muscle functional unit" for publication in PLOS Biology. I have now obtained advice from three of the original reviewers and have discussed their comments with the Academic Editor. 

We're delighted to let you know that we're now editorially satisfied with your manuscript. However before we can formally accept your paper and consider it "in press", we also need to ensure that your article conforms to our guidelines. A member of our team will be in touch shortly with a set of requests. As we can't proceed until these requirements are met, your swift response will help prevent delays to publication. Please also make sure to address the data and other policy-related requests noted at the end of this email.

*Copyediting*

*Published Peer Review History*

*Early Version*

*Submitting Your Revision*

Sincerely,

Ines

--

Ines Alvarez-Garcia, PhD,

Senior Editor,

ialvarez-garcia@plos.org,

PLOS Biology

Fig. 3K; Fig. 4I and Fig. 5C

Reviewers’ comments

Rev. 2:

The authors have extensively addressed the comments from the reviewers.

Rev. 3:

The authors have addressed all points raised in my original review.

Rev. 4:

The authors have fully addressed my previous comments and I have no further concerns.

---

## [Editor Report · Decision Letter 3]

1 Oct 2020

Dear Dr Comai,

On behalf of my colleagues and the Academic Editor, Simon M Hughes, I am pleased to inform you that we will be delighted to publish your Research Article in PLOS Biology. 

Early Version

PRESS 

Kind regards,

Alice Musson

Publishing Editor, 

PLOS Biology

on behalf of

Ines Alvarez-Garcia,

Senior Editor

PLOS Biology